## PROCEEDINGS A

applied mathematics, mathematical modelling, biochemistry

uniform polyhedra, Platonic group, near-miss cages, protein cage, nano-cage, capsid

**Author for correspondence:**
Bernard M. A. G. Piette
e-mail: b.m.a.g.piette@durham.ac.uk

# Characterization of near-miss connectivity-invariant homogeneous convex polyhedral cages

Bernard M. A. G. Piette[1], Agnieszka Kowalczyk[2,3] and Jonathan G. Heddle[2]

[1]Department of Mathematical Sciences, Durham University, South Road, Durham DH1 3LE, UK
[2]Malopolska Centre of Biotechnology, Jagiellonian University, Gronostajowa 7A, Krakow 30-387, Poland
[3]Faculty of Mathematics and Computer Science, Jagiellonian University, Lojasiewicza 6, Krakow 30-348, Poland

BMAGP, 0000-0001-9777-603X

Following the discovery of a nearly symmetric protein cage, we introduce the new mathematical concept of a near-miss polyhedral cage (p-cage) as an assembly of nearly regular polygons with holes between them. We then introduce the concept of the connectivity-invariant p-cage and show that they are related to the symmetry of uniform polyhedra. We use this relation, combined with a numerical optimization method, to characterize some classes of near-miss connectivity-invariant p-cages with a deformation below 10% and faces with up to 17 edges.

## 1. Introduction

Recently, a structure referred to as a TRAP-cage, made out of 24 nearly regular hendecagons, was engineered from TRAP (trp RNA-binding attenuation protein) [1–4]. The structure is such that each hendecagon has five neighbours with which it shares an edge. This leaves six edges per face which define the boundary of 38 holes. Thirty-two of the holes are triangles whereas the remaining six are in between four hendecagons, each contributing two of their edges to them (figure 1).

*(a)* *(b)* *(c)*

**Figure 1.** (*a*) The structure of a TRAP-cage as determined using cryo-electron microscopy [3]. The cage is shown in surface view with each TRAP ring (made from 11 identical protein monomers) coloured a different colour. (*b*) Polyhedral representation of the TRAP-cage: 24 hendecagons, 32 triangular holes and six non-planar holes. (*c*) As in (*b*), but viewed from an axis centred in between two triangular holes. (Online version in colour.)

Such a regular structure is mathematically impossible but can be realized if the edge lengths and angles of the polygons are deformed by as little as 0.5%. This makes such a structure look totally regular and symmetric, although in reality it is only nearly so [3].

Given the huge library of diverse protein structures known, there may be a number of proteins from which it would be useful to build cage-like structures but which may have been overlooked owing to the mathematical 'impossibility' of them forming a regular cage. This raises the question of whether other protein cages with such nearly symmetrical geometries could be made. We start by defining the new concept of a polyhedral cage [5], referred to as a p-cage, as an assembly of (nearly) regular planar polygons, which we also call faces, with holes in between them. The holes can have any shape and do not have to be planar. From a biochemical point of view, the faces will be made out of proteins, while the holes can be empty or can be used to attach particular molecules of choice.

The mathematical concept of a regular shape is not new. Uniform polyhedra are assemblies of regular polygons such that all the vertices are equivalent. It was Kepler who showed [6] that the only convex uniform polyhedra are the Platonic and Archimedean solids, as well as prisms and antiprisms. The non-convex versions were described by Coxeter *et al.* [7]. Johnson [8] generalized the concept to strictly convex assemblies of regular polygons without requiring any equivalence between the vertices. Johnson [8] also listed the 92 so-called Johnson solids and Zalgaller [9] later proved that the list was complete. This was further extended [10] to near-miss Johnson solids as strictly convex assemblies of nearly regular polygons. Because of the holes, p-cages are neither proper solids nor polytopes and so do not fall within the already classified polyhedra. What we are defining is a further generalization of convex regular assemblies of (nearly) regular polygons by allowing holes, of any shape, in the structure.

## (a) Definition

We define a polyhedral cage as an assembly of planar polygons, which we also refer to as faces, separated by holes which do not need to be either regular or even planar. The edges of the p-cage faces are either *shared* with another face or with a hole. Of two adjacent edges at least one of them must be shared with a hole. Moreover, we impose that two adjacent faces do not share more than one edge with each other and that each face must have at least three neighbour faces. Examples of p-cages are presented in figures 1 and 2.

A p-cage is defined as regular if all of its faces are regular planar polygons, but the bionanometric cages such as the TRAP-cage are made out of strictly planar but nearly regular polygonal faces. We refer to these as near-miss p-cages. The amount of deformation is subjective, but in this paper we will consider edge lengths and angles between edges differing by up to 10% from the regular polygons. This is motivated by the fact that p-cages with such a deformation

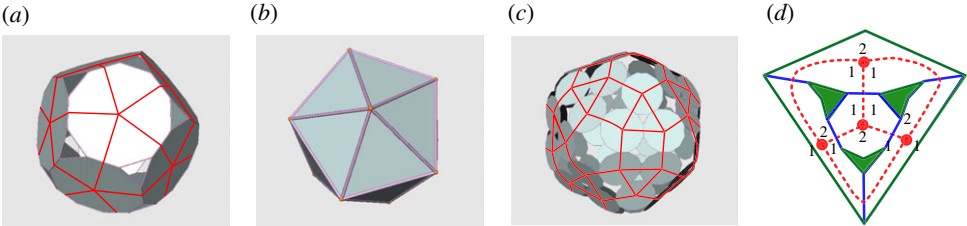

**Figure 2.** (*a*) A p-cage made out of $N = 12$ decagons ($P = 10$). Each hole is made out of $Q = 3$ faces, each contributing one edge to the hole. The red line, drawn on the p-cage faces for clarity, illustrates the construction of the hole-polyhedron. (*b*) The hole-polyhedron, an icosahedron, for the p-cage in (*a*). (*c*) A p-cage made out of $N = 60$ dodecagons ($P = 12$) with three types of holes: $Q = 3$, $Q = 4$ and $Q = 5$. Each face contributes, respectively, one, two and three edges to each of these holes. (*d*) Planar projections of four heptagons placed on the vertices (large dots) of a tetrahedron (dotted lines). The holes are painted except the one underneath the projection. Each hole is made out of two edges from one face and one edge from two different faces, as indicated by the numbers around each edge. (Online version in colour.)

are noticeably irregular to the naked eye, but not excessively so. It is also motivated by the fact that nano-cages with deformation close to or exceeding 2% are known to be likely (AP Biela 2019, personal communication) [11].

The hull of a p-cage is obtained by extending all the faces to infinite planes and considering the interior of all the resulting intersecting half-spaces. The hull will always be a polyhedron, but the faces will usually be irregular. We define a p-cage as convex if its hull is convex.

The p-cage graph is the graph generated from the edges of the p-cage. Some of the nodes will belong to three edges and be part of two faces and one hole while the others will only belong to two edges and be part of one face and one hole. A homogeneous p-cage is defined as a p-cage for which all the faces are polygons with the same number of edges. A homogeneous p-cage is said to be connectivity invariant if all the faces are indistinguishable from their connectivity; in other words, if for any pair of faces $A$ and $B$ there is an automorphism of the p-cage graph onto itself that maps the vertices of face $A$ onto the vertices of face $B$, such that the connecting edges are mapped to connecting edges and so that hole edges are also mapped to hole edges. The faces are assumed to be isomorphic but not isometric. For nanobiotechnological motivations, in this paper, we do not consider the p-cages for which one must involve a reflection to achieve the connectivity invariance. We are also only interested in convex p-cages.

Connectivity invariance is introduced because bionano-cages with that property will be able to assemble randomly/thermally following a larger number of possible assembly paths and, as a result, are more likely to be generated experimentally.

In what follows, we will use the following notations: $N$ will stand for the total number of faces of a p-cage while $P$ will refer to the number of edges of each face. As we will only consider homogeneous cages herein, $P$ will be defined for each p-cage. We will also denote as $Q$ the number of faces surrounding a given hole. In general, p-cages will have holes made out of a different number of faces and so there will be different values of $Q$ for a given cage (figures 1 and 2.)

The paper is organized as follows. We start by defining the dual of the p-cage, which we call the hole-polyhedron. We then show that each p-cage can be constructed from these hole-polyhedra. For this, we must start by characterizing all the possible ways to distribute face edges to the holes. Finally, we proceed by identifying all the graphs with distributed edges corresponding to connectivity-invariant p-cages (either regular or near-miss) for $P$ ranging between 6 and 17.

We then describe a method that will be used to construct all the connectivity-invariant p-cages as well as some geometric constraints used to rule out p-cages with deformation exceeding our chosen threshold. To achieve this, we describe a quality function which measures the non-regularity of p-cages and which one must minimize to find the most regular configuration

for the convex near-miss p-cages. In the final section, we will describe the cages we have found, focusing our attention on the most regular ones.

## (b) Hole-polyhedron

If we join the centre of each face of a p-cage to the centres of the faces that share one edge with it and project this skeleton on a plane, we obtain a planar graph which can also be seen as the three-dimensional (3D) graph of a polyhedron (the faces of the 3D graph will not necessarily be planar, but the vertices of the graph can be projected onto a plane so that none of the edges cross each other.). This is effectively the dual of the p-cage, but as the faces of that polyhedron bear information about the holes of the p-cage, we call it a hole-polyhedron. The $Q$-gonal faces of the hole-polyhedron do surround the p-cage holes made out of $Q$ faces and capture the connectivity of the p-cages: the vertices, edges and faces of the hole-polyhedron correspond, respectively, to the faces, shared edges and holes of the p-cage (figure 2).

To create p-cages, we can proceed backwards and consider any planar graph as a hole-polyhedron, placing a polygon on each vertex (figure 2). One must then distribute the edges of each face between the adjacent faces and holes. The edges of the hole-polyhedron specify which polygons are adjacent on the p-cage. When we add a $P$-gon on the vertex of degree $E_h$, we must join $E_h$ of its edges to neighbour faces, distributing the remaining $P - E_h$ edges, the hole-edges, between the holes. This step is not unique as the hole-edges can be distributed in several ways. For example, an octagon with three neighbours can contribute one, two and two edges to the three holes or one, one and three. This will result in a number of different p-cages. If the polygons are regular and identical, the p-cage will be regular. If the polygons are slightly deformed regular polygons, the p-cage will be near-miss.

By considering all the planar graphs of interest and all the possible repartitions of the edges, we will obtain all the possible p-cage connectivities. While this is very simple conceptually, the number of possibilities grows very quickly with the size of the hole-polyhedron and with $P$, so much so that it very quickly becomes intractable [12,13].

From a bionanotechnology point of view, the most relevant p-cages are the ones where all the faces have the same number of edges and play an equivalent role. This is because such cages, by biological necessity, are built from multiple identical building blocks with identical amino acid sequences and held together by specific interactions between particular amino acids. So we have decided to restrict ourselves to homogeneous connectivity-invariant p-cages. This reduces considerably the number of possible p-cage geometries.

Note that the hole-polyhedra of homogeneous connectivity-invariant p-cages correspond to regular planar graphs. Moreover, the connectivity invariance of the p-cage implies that the hole-polyhedron is vertex transitive or, in other words, a Cayley graph. These graphs were classified by Maschke [14] and are the Platonic and Archimedean solids as well as the uniform prisms and uniform antiprisms. As we disregard connectivity invariance via reflection, we must exclude the truncated cuboctahedron and truncated icosidodecahedron.

## (c) Repartition of edges on the p-cage holes

The next task we must perform is to determine all the connectivity-invariance-preserving ways to distribute the face edges of the p-cage between the holes. For a p-cage hole with $E_h$ edges per vertex and $P$-gonal faces, this is equivalent to distributing the strictly positive numbers $a_i$, $i = 1, \ldots, E_h$, around each hole-polyhedron vertex,

$$P = E_h + \sum_{i=1}^{E_h} a_i, \tag{1.1}$$

in such a way that the p-cage is connectivity invariant; in other words, such that, for any two vertices of the hole-polyhedron, there is at least one automorphism of the hole-polyhedron that maps the first vertex onto the second and that preserves the distribution of the $a_i$.

**Table 1.** Symbols for convex uniform solids.

| solid | SYM | solid | SYM |
|---|---|---|---|
| triangular prism | tp | tetrahedron | Pte |
| square prism (cube) | Pcu | octahedron | Poc |
| pentagonal prism | pp | dodecahedron | Pdo |
| hexagonal prism | hp | icosahedron | Pic |
| heptagonal prism | 7p | truncated cube | Atc |
| octagonal prism | 8p | truncated tetrahedron | Att |
| nonagonal prism | 9p | truncated octahedron | Ato |
| decagonal prism | 10p | truncated dodecahedron | Atd |
| triangular antiprism | ta | truncated icosahedron | Ati |
| square antiprism | sa | snub cube | Asc |
| pentagonal antiprism | pa | snub dodecahedron | Asd |
| hexagonal antiprism | ha | cuboctahedron | Aco |
| heptagonal antiprism | 7a | rhombicuboctahedron | Arco |
| octagonal antiprism | 8a | rhombicosidodecahedron | Arcd |
| nonagonal antiprism | 9a | icosidodecahedron | Aid |
| decagonal antiprism | 10a | | |

We will now perform the construction graphically using the letters 'a', 'b', 'c', 'd' and 'e' instead of $a_i$. As this labelling will later be used to name the p-cages, we have adopted the following convention to decide on which face the 'a' is placed for the first and arbitrary set of labels. For prisms and antiprisms, the label 'a' is placed on the base polygon. For Platonic solids, it does not matter as all the faces are identical, while for Archimedean solids we place the 'a' on the face with the smallest number of edges except for the snub cube and the snub dodecahedron because this could lead to an ambiguity as there are two types of triangles for these solids. We thus place the label 'a', respectively, on the square and the pentagon for these solids. The other labels are then placed anti-clockwise around the vertices in alphabetical order.

To identify the p-cages unambiguously, we have adopted a notation made out of three parts: `SYM_PN_QI`, where `PN` is the letter P followed by the number of edges of the p-cage faces; `SYM` refers to a symbol, listed in table 1, used to specify the hole-polyhedron from which the p-cage is made; finally, `QI` refers to the diagram values of 'a', 'b', 'c', 'd' and 'e' separated by the symbol '_'. For example, the p-cage in figure 2a is called `Pic_P10_1_1_1_1_1` because its hole-polyhedron is the icosahedron, its faces have 10 edges and each face contributes five times a single edge to the holes (a=b=c=d=e=1 in figure 2a).

We now consider each regular solid in turn, starting with the prisms but excluding the cube that has a higher symmetry. By connectivity invariance, all the corners of the base of the prism must be identical, as they can only be mapped between themselves via a rotation of the base, and we label them 'a'. The corners of the squares, forming the sides of the prism, can then have a different number of edges, labelled 'b' and 'c' but diagonally opposite corners must have the same value. This is shown graphically in figure 3.

Similarly for antiprisms, the corners of the base of the prism must be identical. The corners of the triangles can then have a different number of edges, labelled 'b', 'c' and 'd', as shown i figure 3.

Before we consider all the Platonic and Archimedean solids, we consider the possible configurations for a triangle, as shown in figure 4, as this will be a recurrent structure which

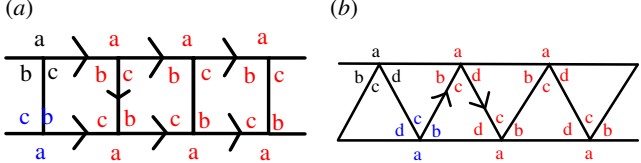

**Figure 3.** Repartition of edges: (*a*) on a prism; (*b*) on an antiprism. The arrows show the order in which the labelling is performed: one starts with the arbitrary labels (top left), moves on to the next ones (bottom left) and then infers the following ones. (Online version in colour.)

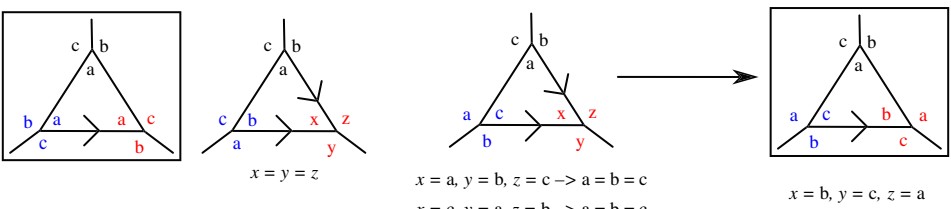

**Figure 4.** Repartition of edges on an $E_h = 3$ triangle. The non-trivial connectivity-invariant configurations are surrounded by a box. (Online version in colour.)

we will use several times. We start by placing the labels 'a', 'b' and 'c' around the top vertex. We then place the three labels on the bottom left vertex in the three possible positions and consider each case in turn. We then try to impose the connectivity invariance on the other vertices. For the first case, we see that the pair 'c', 'a' faces the pair 'a', 'b' and this means that on the third vertex 'a', 'b' must face 'c', 'a' from the second vertex. We also notice that the connectivity invariance is satisfied on the third edge so this configuration is invariant. For the second case, there is no connectivity invariance we can apply, so we try all three possibilities on the third vertex and, in all three cases, we see that the only possible configuration is the trivial one where 'a'='b'='c'. For the third case, we only have one possible configuration. We can then conclude that the only possible configurations are the ones for which the three corners of the triangle have either the same label or three different ones, in which case they are ordered clockwise.

We can do the same construction for a triangle with vertices connected to four other vertices (figure 5). In this case, we also see that the vertices of the triangle must be either all identical or all different. When different they must be ordered clockwise and there are three different ways to do this as one of the labels must be missed out (except 'a', which is fixed.)

Finally, we also consider a square with vertices connected to three other vertices (figure 6). In this case, we see that the four corners of the square must either have the same label or have two different ones with identical labels for diagonally opposite corners.

We will now construct the different repartitions for the Platonic solids. For the tetrahedron, $E_h = 3$ and it is not possible to fit three 'a' on one face as this leads to contradiction unless all three labels are identical. One must then have three different labels on each face and one obtains the diagram shown in figure 7. For the cube, $E_h = 3$ and we can place four 'a' on each face or alternating 'b' and 'c'. In both cases, we obtain the diagram shown in figure 7. For the octahedron, we can place three 'a' on the same face or three different labels in clockwise order. When trying all possible combinations, one obtains two different diagrams. In the first one, named Poc1, 2 opposite faces have three 'a', while the others have the remaining three labels. In the second, named Poc2, each face has only 'a' or only 'b' in such a way that similar faces do not share an edge (figure 7) (see electronic supplementary material).

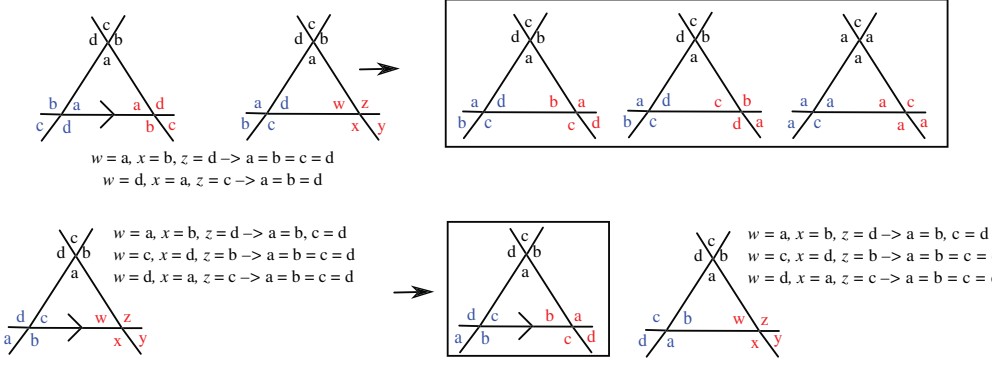

**Figure 5.** Repartition of edges on an $E_h = 4$ triangle. The non-trivial connectivity-invariant configurations are surrounded by a box. (Online version in colour.)

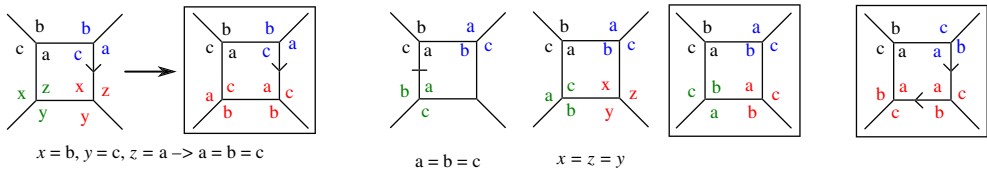

**Figure 6.** Repartition of edges on an $E_h = 3$ square. The non-trivial connectivity-invariant configurations are surrounded by a box. (Online version in colour.)

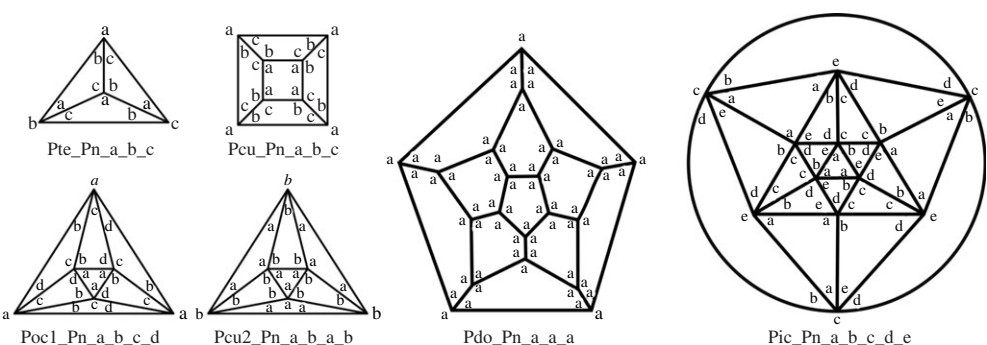

**Figure 7.** Repartition of hole-edges on the Platonic hole-polyhedra.

For the dodecahedron, the only diagram which ensures connectivity invariance for the p-cages is when all the corners have the same label, so we only have `Pdo_Pn_a_a_a`. For the icosahedron, there is only one possible diagram, modulo some equivalence and it is shown in figure 7 (see electronic supplementary material for the proof).

We can now construct the connectivity invariance diagrams for the Archimedean solids. The truncated tetrahedron, truncated cube, truncated dodecahedron, rhombicuboctahedron and rhombicosidodecahedron all have one triangle per vertex. By connectivity invariance, this implies that the corners of triangles must all be 'a' and that the other labels are uniquely distributed.

The truncated octahedron and snub cube have one square per vertex, implying that these faces must have 'a' in each corner and that this determines the position of the other labels. The same is

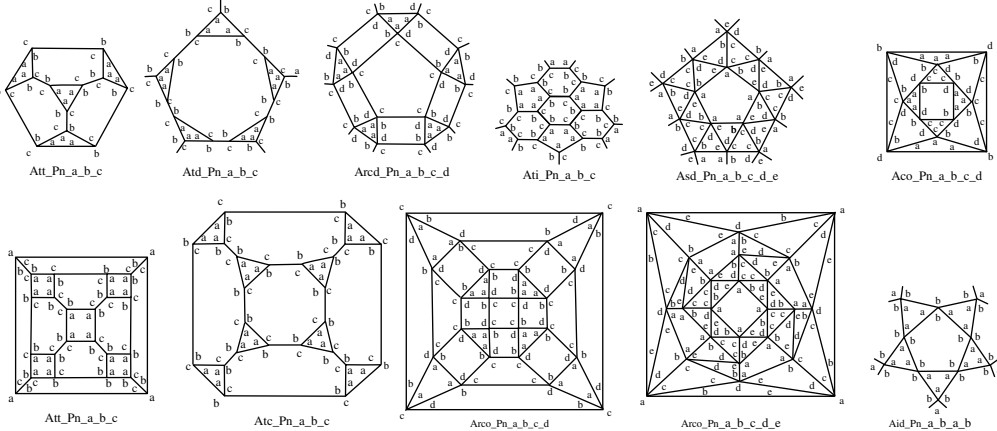

**Figure 8.** Repartition of hole-edges on the truncated tetrahedron, truncated dodecahedron, rhombicosidodecahedron, truncated icosahedron, snub dodecahedron, cube octahedron, truncated octahedron, truncated cube, rhombicuboctahedron, snub cube and icosidodecahedron. For the large solids, we only present a section of the diagram.

true for the truncated icosahedron and snub dodecahedron, which have one pentagon per vertex. This is shown in figure 8.

The vertices of the cuboctahedron are adjacent to two triangles and two squares. We can thus place three 'a' on one of the triangles and as we do so the triangles on the opposite side of the vertex must be 'c' and the diagram is completely determined (figure 8). One could also start with three different labels clockwise on one of the triangles, but this is not possible without breaking the connectivity invariance unless the labels are all identical.

The vertices of the icosidodecahedron are adjacent to two triangles and two pentagons. We can thus place three 'a' on one of the triangles and as we apply the connectivity invariance rule we find that 'a'='c' and 'b'='d'. So all the triangles must be filled with 'a' and all the pentagons with 'b'. It is not possible to fill a triangle with three different labels without breaking the connectivity invariance.

The vertices of the truncated cuboctahedron and truncated icosidodecahedron are not invariant if one excludes reflections. This is easily seen by noticing that the rotation symmetry around the axis going through an n-gonal face is a rotation of $4\pi/n$ and not $2\pi/n$. As a result, it is not possible to generate connectivity-invariant p-cages from these two Archimedean solids.

## 2. Constraints on holes

While in principle we could consider all the possible distributions of edges on the holes, some of them lead to configurations which cannot correspond to a p-cage, or ones for which the deformation would be too large. By deformation, we mean angles different from the angle of the regular polygon or edge lengths different from a reference length, which we will ultimately set to 1. To make the problem tractable, we start by deriving a set of constraints on the holes and their edges to guarantee face deformations below a set threshold.

We start by defining the p-cage sub-face as the polygon, usually irregular, made out of the edges shared with other p-cage faces and completed by replacing the edges contributing to the holes by a straight line joining the two exterior vertices (figure 9a). These sub-faces will be hexagons, octagons and decagons, respectively, for faces with three, four and five neighbours. The p-cage sub-faces generate a $Q$-gon, which is usually not flat, around any hole made out of $Q$ faces. We call it the sub-face hole.

If we take the faces surrounding a hole and *disjoin* two of the adjacent faces, one will be able to flatten the structure onto a plane and the two faces that have been severed will not overlap

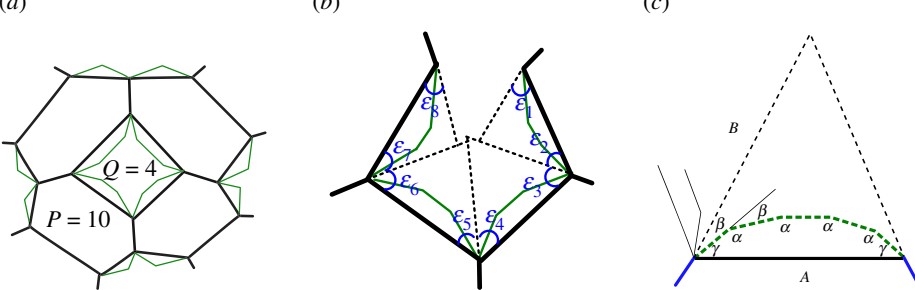

**Figure 9.** (*a*) Sub-face (black) for a p-cage with $P = 10$ and a hole with $Q = 4$. (*b*) Structure of the flattened sub-faces hole. (*c*) Close-up view of face-edges contributing to a hole and depicting the angles $\alpha$ between the face hole edges (segmented dotted lines) and the angles $\gamma$ between the face hole edges and the sub-face hole edge (black bold line). Note $\epsilon_i = \gamma_i + \beta_i$. (Online version in colour.)

(figure 9*b*). This means that the edges of the sub-face holes will not be intersecting and will not close into a polygon. This in turn implies that, to form a convex p-cage, we must impose that the sum of the angles of the sub-faces hole, $\epsilon_i$ in figure 9*b*, once projected onto a plane must be greater than the sum of the angles of a $Q$-gon.

As illustrated in figure 9*c*, we call $\beta$ the angle between two adjacent edges of a polygon and $\alpha = \pi - \beta$ the inner angle between the two edges. $\gamma$ denotes the angle between the edge of a sub-face hole and the p-cage edge adjacent to it.

Following the notation of figure 9 and using the index $i$ to label the faces, the sum of the $\epsilon_i$ is equal to twice the sum of the $\beta_i + \gamma_i$ and we thus have

$$\sum_{i=1}^{Q} 2(\beta_i + \gamma_i) \geq Q\,\pi\left(1 - \frac{2}{Q}\right), \tag{2.1}$$

where we have used the fact that the sum of the inside angles of an $S$-gon is $S\pi(1 - 2/S)$. In what follows, we will use the index 0 to denote the angles of the regular polygons/faces, $\beta_{i,0} = 2\pi/P$, and write $\beta_i = \beta_{i,0}(1 + \kappa_i)$ for a non-regular polygon, where $\kappa_i$ is a deformation factor which can differ between faces, hence the index $i$. We then have $\alpha_{i,0} = \pi(1 - 2/P)$ and $\alpha_i = \pi - \beta_i = \pi(P - 2 - 2\kappa_i)/P$. If the face $i$ contributes $q_i$ edges to the hole, we must then have

$$2\gamma_i + (q_i - 1)\alpha_i = \pi(q_i - 1) \quad \text{and} \quad \gamma_{i,0} = \pi\frac{q_i - 1}{P} = \beta_{i,0}\frac{q_i - 1}{2}. \tag{2.2}$$

Substituting the expression for $\alpha_i$, we obtain $\gamma_i = \gamma_{i,0}(1 + \kappa_i)$ and the constraint (2.1) becomes

$$Q\pi\left(1 - \frac{2}{Q}\right) \leq 2\sum_i (\beta_{i,0} + \gamma_{i,0})(1 + \kappa_i) \leq (1 + \max_i(\kappa_i))2\sum_i(\beta_{i,0} + \gamma_{i,0}). \tag{2.3}$$

In the construction of the p-cages, it is the deformation of the angle $\alpha$ which we use as the deformation parameter. Denoting it $\kappa_\alpha$, we have $\alpha_i = \alpha_{i,0}(1 - \kappa_{\alpha,i})$, where $\kappa_i = \kappa_{\alpha,i}(P - 2)/2$. Then, as $\beta_{i,0} + \gamma_{i,0} = \pi(1 + q_i)/P$, we can rewrite (2.1) and (2.3) as

$$P \leq 2\frac{\sum_{i=1}^{Q}(q_i + 1)}{Q - 2}\left(1 + \max_i(\kappa_{\alpha,i})\frac{P - 2}{2}\right). \tag{2.4}$$

The second constraint we can derive is that the edge length $A$ of the sub-faces hole must be smaller than the sum of the edge lengths of the other sub-faces hole contributing to the same hole,

$$A_j \leq \sum_{i=1,i\neq j}^{Q} A_i. \tag{2.5}$$

When the equality sign holds, one can join the faces together by deforming the polygons with the smallest contribution to the hole so that their $\gamma$ is 0. We can assume that the extreme configuration is one where all the edges and angles are all stretched or contracted to the maximum amount. So, when evaluating (2.5) we must assume $A_j$ on the left must be deformed to become as small as possible while $A_i$ on the right-hand side are to be as large as possible. We need to perform the test taking each $A_i$ of the hole on the left-hand side and rule out any cage for which the test fails.

If $q_k$ is even, we have

$$A_{k,\text{even}} = 2\mathcal{A} \sum_{i=0}^{q_k/2-1} \cos(\gamma_k - i\beta_k) = 2\mathcal{A} \sum_{i=0}^{q_k/2-1} \cos(\pi\, K_{k,i} \frac{q_k - 1 - 2i}{P}), \tag{2.6}$$

where $\mathcal{A}$ is a length scale where we take $K_{k,i} = 1 + \kappa_i$ or $K_{k,i} = 1/(1 + \kappa_i)$ depending on if we want to majorate or minorate $A_i$. If $q_k$ is odd, we have

$$A_{k,\text{odd}} = \mathcal{A} \left( 1 + 2 \left( \sum_{i=0}^{(q_k-3)/2} \cos(\gamma_k - i\beta_k) \right) \right)$$

$$= \mathcal{A} \left( 1 + 2 \left( \sum_{i=0}^{(q_k-3)/2} \cos(\pi\, K_{k,i} \frac{q_k - 1 - 2i}{P}) \right) \right). \tag{2.7}$$

Note that, for $q_k = 1$, $A_{k,\text{odd}} = \mathcal{A}$. To help satisfy (2.5) we can stretch the shorter lengths by a factor up to $1 + \kappa_t$, where $\kappa_t$ is the threshold deformation factor, and shorten the longest one by the same amount, so (2.5) becomes

$$A_j \leq \frac{1 + \kappa_t}{1 - \kappa_t} \sum_{i=1, i\neq j}^{Q} A_i, \tag{2.8}$$

where we compute $A_j$ using $K_{k,j} = 1/(1 + \kappa_t)$ and the $A_i$ on the right-hand side using $K_{k,i} = 1 + \kappa_t$.

The conditions (2.4) and (2.8) allow one to rule out many possible p-cage configurations. For the configurations which fulfil those two conditions, we must construct the corresponding p-cages and deform the polygonal faces until one obtains a convex p-cage with planar faces.

This can be achieved by using a computer program where the vertices of the polygons are moved so that all the necessary conditions for the p-cage are obtained. Some p-cages differ only by a chiral transformation. In that case, we have only kept one of them. For $P$ ranging from 6 to 17 and considering only prisms and antiprisms with bases ranging from triangles to decagons as well as the Platonic and 11 of the Archimedean solids, we found 5743 potential p-cage configurations satisfying condition (2.8). The list of all these configurations is given in the electronic supplementary material.

Many of these cage configurations will have angles and edge length deformations larger than 10%. To discard them, we need to realize each of these cages geometrically and minimize the amount of deformation for the angles and the edges of the faces. This is achieved by defining and then minimizing numerically a functional which measures the amount of deformation of the p-cage faces. One starts from an approximate position for the vertices of all the faces and then randomly displaces them using a Metropolis algorithm to optimize the functional, which we will now define.

## 3. Deformation functional

To optimize the regularity of the faces of a convex p-cage we define a functional which is the sum of five terms. The first two measure, respectively, the amount of deformation of the face edge lengths and the face angles. The third measures the non-planarity of the faces, while the last two measure the convexity of, respectively, the faces and the p-cage. We need to impose planarity as a constraint, rather than geometrically, because the vertices of the p-cage faces are the degree of freedom we need to optimize. Each of these five terms is then multiplied by a weight factor as

explained below. There is no reason to assume that all the p-cage faces will be deformed the same way and become identical. In our optimization, we thus assume that the vertices of the p-cage are independent parameters.

Before we define each of these functionals, we need to define a few quantities. First of all, we call *node* the vertices of the p-cage faces and we denote by $\mathcal{N}_N$ the total number of the nodes for the p-cage (each counted only once). To keep our notation compact, we use $P\%n$ as the operator $P$ modulo $n$ for any pair of integers $P$ and $n$. We then denote by $n_{f,i}$ the coordinates of node $i = 0, \ldots, P - 1$, of face $f$, while $s_{f,i} = n_{f,(i+1)\%P} - n_{f,i}$ corresponds to the vector spanning the edge between node $i$ and node $i + 1$ where the vectors are oriented so that they rotate anti-clockwise when looking at the face from outside the p-cage. As a result, the angle at node $i$ of face $f$ is

$$\alpha_{f,i} = \arccos\left(-\frac{(s_{f,i} \cdot s_{f,(i+1)\%P})}{|s_{f,i}|\,|s_{f,(i+1)\%P}|}\right). \tag{3.1}$$

We define the centre of the p-cage, $O$, and the centre of the $P$-gonal face $f$, $C_f$, as

$$O = \frac{1}{\mathcal{N}_N} \sum_{\substack{i,f \\ \text{count each} \\ \text{node once}}} n_{f,i} \quad \text{and} \quad C_f = \frac{1}{P} \sum_{i=0}^{P-1} n_{f,i}. \tag{3.2}$$

The centre of a face relative to the centre of the p-cage is then $F_f = C_f - O$. We now introduce the following vectors:

$$v_{f,i} = n_{f,i} - C_f, \quad M_{f,i} = (n_{f,i} + n_{f,(i+1)\%P})/2 \quad \text{and} \quad d_{f,i} = M_{f,i} - C_f. \tag{3.3}$$

$v_{f,i}$ are the local vertex coordinates relative to the centre of the face, i.e. the vector joining the centre of face $f$ and node $n_{f,i}$. $M_{f,i}$ is the vector position of the centre of the edge linking node $i$ and node $i + 1$ and $d_{f,i}$ is the vector from the centre of a face to the centre of an edge.

We now define a facelet as the triangle spanned by two adjacent vectors $v_{f,i}$. Its normal vector $w_{f,i}$ and the area vector $W_f$ of face $f$ are given, respectively, by

$$w_{f,i} = \frac{1}{2}(v_{f,i} \times v_{f,(i+1)\%P}) \quad \text{and} \quad W_f = \sum_{\text{nodes}} w_{f,i}. \tag{3.4}$$

For flat faces, $W_f$ is a vector perpendicular to the face and of length equal to its area. As a result, the vector orthonormal to the face $f$ is

$$\widehat{W}_f = \frac{W_f}{|W_f|}. \tag{3.5}$$

We now define the different terms for a functional which we will use to minimize the deformation of the p-cage faces while ensuring face planarity as well as face and p-cage convexity.

## (a) Face regularity

The first constraint we want to impose is that the lengths of the edges of the faces are as close as possible to a reference length $L_f$ and we thus define the following least-squares quality function:

$$\mathcal{Q}_{\text{length}} = \sum_{f \in \text{faces}} \sum_{i \in \text{edges}} \left(\frac{l_{f,i} - L_f}{L_f}\right)^2. \tag{3.6}$$

Similarly, we want to impose that the angles $\alpha_{f,i}$ of the faces are as close as possible to the angle of a regular polygon, which, for a polygon with $P$ edges, is given by $\pi(1 - 2/P)$. We thus define

the quality function

$$Q_{\text{angle}} = \sum_{f \in \text{faces}} \sum_{i \in \text{corners}} \left( \alpha_{f,i} - \pi \left( 1 - \frac{2}{P} \right) \right)^2.$$ (3.7)

We have chosen these two functions so that they carry similar weight. This can be justified by considering an isosceles right triangle and deforming it so that the long edge, $L$, is elongated by a small amount. This can be achieved by either keeping the two smaller edges $l$ at the same length and changing the right angle $\theta$, or by keeping the right angle and elongating one of the smaller edges. In the second case, we have $l_1 = l(1 + \epsilon)$, $l_2 = l$ and $\theta = \pi/2$. Then $L^2 \approx 2l^2(1 + \epsilon)$. In the first case, we have $L = 2l \sin((\pi/4) + (\delta/2)) \approx 2l(\sin(\pi/4) + \cos(\pi/4)\delta/2) = \sqrt{2}l(1 + \delta/2)$, where $\delta$ and $\epsilon$ are small deformation parameters.

Comparing the two cases, we have

$$l\sqrt{2(1 + \epsilon)} \approx \sqrt{2}l \left( 1 + \frac{\delta}{2} \right) \quad \text{or} \quad 1 + \frac{\epsilon}{2} \approx 1 + \frac{\delta}{2},$$ (3.8)

implying that $(\Delta l)^2/l^2 = (\Delta \theta)^2$ if $\theta$ is measured in radians. We indeed found that for many p-cages the most regular configurations were obtained when these two functions have roughly the same weight.

## (b) Face planarity and convexity

We must also impose that the faces are planar. We need to impose the planarity constraint numerically because imposing it analytically would involve solving a large number of algebraic equations, which would make the minimization algorithm computationally far too slow. We can do this by imposing that all the facelet vectors $w_{f,i}$ are parallel to each other and parallel to the face normal vector $\widehat{W}_f$ and define the quality function

$$Q_{\text{planar}} = \sum_{f \in \text{faces}} \sum_{i \in \text{edges}} w_{f,i}^t (1 - \widehat{W}_f \widehat{W}_f^t) w_{f,i},$$ (3.9)

which corresponds to the sum of the squares of the projected lengths of the facelet vectors onto the face plane. This evaluates to 0 if the face is planar.

We must also impose that each face is convex and, as the edge vectors $s_{f,i}$ are rotating anti-clockwise, the vector $s_{f,i} \times s_{f,(i+1)\%P}$ must point towards the outside of the face and so we must have $(s_{f,(i+1)\%P} \times s_{f,i}).F_f > 0$. We can then use the following quality function:

$$Q_{\text{ConvFace}} = \sum_{f \in \text{faces}} \sum_{i \in \text{edges}} H((s_{f,(i+1)\%P} \times s_{f,i}).F_f),$$ (3.10)

where $H(\cdot)$ is the Heaviside function.

## (c) Convexity of a p-cage

We must finally impose the condition that the p-cage is convex. As we will optimize the quality function using the Metropolis algorithm, we must use an expression which depends on as few points as possible so as to make the algorithm as fast as possible. To achieve this, we impose that two adjacent faces, i.e. sharing an edge, must be *bent* towards the centre of the cage. Mathematically, this implies that if the faces $f$ and $f'$ are adjacent and touching at their respective edges $i$ and $i'$, the sum of the two vectors $d_{f,i}$, defined in (3.3), must be pointing away from the

centre of the cage; in other words,

$$\left( \frac{d_{f,i}}{|d_{f,i}|} + \frac{d_{f',i'}}{|d_{f',i'}|} \right).(M_{f,i} - O) > 0. \tag{3.11}$$

Note that $M_{f,i} = M_{f',i'}$ and as a quality function we can use

$$\mathcal{Q}_{\text{ConvPol}} = \sum_{f \in \text{faces}} \sum_{i \in \text{edges}} H\left( \left( \frac{d_{f,i}}{|d_{f,i}|} + \frac{d_{f',i'}}{|d_{f',i'}|} \right).(M_{f,i} - O) \right). \tag{3.12}$$

This expression does not strictly impose convexity in all configurations but we found that it works very well in the majority of cases. For some p-cages, we had to use another expression which is more expensive computationally but more rigorous. If we consider the normal unit vectors $\widehat{W}_f$ and $\widehat{W}_{f'}$ of two adjacent faces, the p-cage will be convex if the distance between the base of the two vectors is smaller than the distance between their tips. In other words, we can use as a quality function

$$\mathcal{Q}_{\text{ConvPol}} = \sum_{f \in \text{faces}} \sum_{f' \text{ neighbour of } f} H(|C_f - C_{f'}| - |C_f + \widehat{W}_f - C_{f'} - \widehat{W}_{f'}|). \tag{3.13}$$

## (d) Optimizing functional

Putting all of these functions together, what we have to do is to find p-cages which optimize the function

$$\mathcal{Q} = c_l \mathcal{Q}_{\text{length}} + c_a \mathcal{Q}_{\text{angle}} + c_p \mathcal{Q}_{\text{planar}} + c_{\text{cf}} \mathcal{Q}_{\text{ConvFace}} + c_{\text{cp}} \mathcal{Q}_{\text{ConvPol}}, \tag{3.14}$$

where $c_l, c_a, c_p, c_{\text{cf}}$ and $c_{\text{cp}}$ are weight parameters. For $\mathcal{Q}_{\text{ConvPol}}$, we use (3.12) most of the time except for some cages which prefer to assume a concave configuration and for which using (3.13) works better.

To perform this optimization, we have considered each hole edge repartition for each hole-polyhedron separately, hence fixing the connectivity from the start. We then started from a simple mechanical model of semi-rigid faces connected together by springs. The polygonal faces, connected by springs, were very crudely distributed around a sphere and the system was relaxed to obtain a better estimate of the face positions. We then used a Metropolis algorithm to optimize (3.14) with $L_f = 1$, $c_l = c_a = 1$ and $c_p = 1000$. The convexity parameters were usually set to $c_{\text{cf}} = 100$ and $c_{\text{cp}} = 100$ but in most cases the actual value did not matter as the p-cages were naturally assuming a convex configuration. For some cages $c_{\text{cp}}$ had to be larger or smaller for the optimization to work well.

Once a good configuration was obtained, we used a combined downhill simplex [15] and Metropolis method to relax the configurations for varying $c_l$ and $c_a$ while keeping their sum equal to 2. We used 100 different values spread logarithmically in that interval.

Defining the maximum relative deformation of edge lengths $\Delta_l$ and face angles $\Delta_a$ as

$$\Delta_l = \max_{f,i}(|l_{f,i} - L_f|) \quad \text{and} \quad \Delta_a = \max_{f,i} \left| \frac{\alpha_{f,i} - \pi\left(1 - \frac{2}{P}\right)}{\pi\left(1 - \frac{2}{P}\right)} \right|, \tag{3.15}$$

we then took as the best cage the one which minimizes $\max(\Delta_l, \Delta_a)$. Finally, we used a bisection method, varying $c_l$ and $c_a$ but keeping $c_l + c_a = 2$, to find the cage with the smallest deformation. We then ruled out any cages for which $\max(\Delta_l, \Delta_a) > 0.1$.

When determining the node positions of a p-cage numerically it is possible that some of the faces intersect each other. This occurs mostly for some p-cages derived from prisms. These cage configurations must be ignored. We have thus written a Python program which searches for such intersections using an algorithm derived by Möller [16].

**Table 2.** Number of connectivity-invariant convex p-cages for each polygon.

| P | 6 | 7 | 8 | 9 | 10 | 11 | 12 | 13 | 14 | 15 | 16 | 17 | total |
|---|---|---|---|---|----|----|----|----|----|----|----|----|-------|
| near-miss | 2 | 4 | 12 | 32 | 38 | 63 | 69 | 99 | 117 | 141 | 183 | 228 | 988 |
| regular | 6 | 8 | 15 | 6 | 11 | 11 | 23 | 8 | 11 | 18 | 18 | 14 | 149 |

**Table 3.** Prism and antiprism-based regular connectivity-invariant p-cages. $P = 6$–$17$. $\theta$ is the angle between the p-cage face and the base of the underlying prism. ($P = 18$–$20$ is provided in the electronic supplementary material.)

| Name | $\theta$ (degree) |
|------|-------------------|
| tp_P6_1_1_1 | 70.52877936550934 |
| Pcu_P6_1_1_1 | 54.7356103172454 |
| pp_P6_1_1_1 | 37.37736814064979 |
| hp_P6_1_1_1 | 0 |
| tp_P7_2_1_1 | 82.42774243307761 |
| Pcu_P7_2_1_1 | 76.80632022932768 |
| pp_P7_2_1_1 | 71.69048073229207 |
| hp_P7_2_1_1 | 66.71348712632624 |
| 7p_P7_2_1_1 | 61.70883342181402 |
| 8p_P7_2_1_1 | 56.562399577679 |
| 9p_P7_2_1_1 | 51.163968231716986 |
| 10p_P7_2_1_1 | 45.37519985060495 |
| tp_P8_1_2_2 | 54.73561031724534 |
| tp_P8_1_1_3 | 90.0 |
| tp_P8_3_1_1 | 90.0 |
| Pcu_P8_1_2_2 | 0 |
| Pcu_P8_3_1_1 | 90.0 |
| pp_P8_3_1_1 | 90.0 |
| hp_P8_3_1_1 | 90.0 |
| 7p_P8_3_1_1 | 90.0 |
| 8p_P8_3_1_1 | 90.0 |
| 9p_P8_3_1_1 | 90.0 |
| 10p_P8_3_1_1 | 90.0 |
| tp_P9_2_2_2 | 70.52877936550928 |
| Pcu_P9_2_2_2 | 54.73561031724532 |
| pp_P9_2_2_2 | 37.37736814064964 |
| hp_P9_2_2_2 | 0 |
| tp_P10_1_3_3 | 37.377368140649665 |
| tp_P10_3_2_2 | 79.18768303642828 |
| Pcu_P10_3_2_2 | 71.0392901180775 |
| pp_P10_3_2_2 | 63.43494882292201 |
| hp_P10_3_2_2 | 55.7519065416252 |
| 7p_P10_3_2_2 | 47.56893248471767 |
| 8p_P10_3_2_2 | 38.332421336170036 |
| 9p_P10_3_2_2 | 26.78427512749162 |
| 10p_P10_3_2_2 | 0 |
| tp_P11_2_3_3 | 59.98168165966778 |
| tp_P11_4_2_2 | 85.2383730882915 |
| Pcu_P11_2_3_3 | 29.94500432784639 |
| Pcu_P11_4_2_2 | 81.73346069825418 |

| Name | $\theta$ (degree) |
|------|-------------------|
| pp_P11_4_2_2 | 78.5861762612658 |
| hp_P11_4_2_2 | 75.57980247140252 |
| 7p_P11_4_2_2 | 72.62894391019375 |
| 8p_P11_4_2_2 | 69.68925657161427 |
| 9p_P11_4_2_2 | 66.7323035785031 |
| 10p_P11_4_2_2 | 63.73623869478808 |
| tp_P12_1_4_4 | 0 |
| tp_P12_3_3_3 | 70.52877936550934 |
| tp_P12_5_2_2 | 90.0 |
| Pcu_P12_3_3_3 | 54.7356103172454 |
| Pcu_P12_5_2_2 | 90.0 |
| pp_P12_3_3_3 | 37.37736814064979 |
| pp_P12_5_2_2 | 90.0 |
| hp_P12_3_3_3 | 0 |
| hp_P12_5_2_2 | 90.0 |
| 7p_P12_5_2_2 | 90.0 |
| 8p_P12_5_2_2 | 90.0 |
| 9p_P12_5_2_2 | 90.0 |
| 10p_P12_5_2_2 | 90.0 |
| tp_P13_2_4_4 | 49.330575707965174 |
| tp_P13_4_3_3 | 77.35203942943825 |
| Pcu_P13_4_3_3 | 67.7127678515546 |
| pp_P13_4_3_3 | 58.533979606291815 |
| hp_P13_4_3_3 | 48.93762711332103 |
| 7p_P13_4_3_3 | 38.045584095595515 |
| 8p_P13_4_3_3 | 23.71028385264075 |
| tp_P14_3_4_4 | 62.58569209456164 |
| tp_P14_5_3_3 | 82.42774243307761 |
| Pcu_P14_3_4_4 | 37.11049837738742 |
| Pcu_P14_5_3_3 | 76.80632022932768 |
| pp_P14_5_3_3 | 71.69048073229207 |
| hp_P14_5_3_3 | 66.71348712632624 |
| 7p_P14_5_3_3 | 61.70883342181402 |
| 8p_P14_5_3_3 | 56.562399577679 |
| 9p_P14_5_3_3 | 51.163968231716986 |
| 10p_P14_5_3_3 | 45.37519985060495 |
| tp_P15_2_5_5 | 37.37736814064978 |
| tp_P15_4_4_4 | 70.52877936550934 |
| tp_P15_6_3_3 | 86.52104255798322 |
| Pcu_P15_4_4_4 | 54.7356103172454 |

| Name | $\theta$ (degree) |
|------|-------------------|
| Pcu_P15_6_3_3 | 83.96682796910126 |
| pp_P15_4_4_4 | 37.37736814064979 |
| pp_P15_6_3_3 | 81.68220228396986 |
| hp_P15_4_4_4 | 0 |
| hp_P15_6_3_3 | 79.51105099587654 |
| 7p_P15_6_3_3 | 77.39366209201008 |
| 8p_P15_6_3_3 | 75.30082184052095 |
| 9p_P15_6_3_3 | 73.2155774064996 |
| 10p_P15_6_3_3 | 71.12663520702004 |
| tp_P16_3_5_5 | 54.73561031724534 |
| tp_P16_5_4_4 | 76.16383952074351 |
| tp_P16_7_3_3 | 90 |
| Pcu_P16_5_4_4 | 65.53019947929778 |
| Pcu_P16_7_3_3 | 90 |
| pp_P16_5_4_4 | 55.2416807405721 |
| pp_P16_7_3_3 | 90 |
| hp_P16_5_4_4 | 44.156563020080426 |
| hp_P16_7_3_3 | 90 |
| 7p_P16_5_4_4 | 30.669567363098952 |
| 7p_P16_7_3_3 | 90 |
| 8p_P16_7_3_3 | 90 |
| 9p_P16_7_3_3 | 90 |
| 10p_P16_7_3_3 | 90 |
| tp_P17_2_6_6 | 21.179567726749838 |
| tp_P17_4_5_5 | 64.15139587665604 |
| tp_P17_6_4_4 | 80.54515180736814 |
| Pcu_P17_4_5_5 | 46.96021552181372 |
| Pcu_P17_6_4_4 | 73.46957037520751 |
| pp_P17_6_4_4 | 66.94500928396418 |
| hp_P17_6_4_4 | 60.47449520304961 |
| 7p_P17_6_4_4 | 53.784689840650096 |
| 8p_P17_6_4_4 | 46.61454973762823 |
| 9p_P17_6_4_4 | 38.58123808904467 |
| 10p_P17_6_4_4 | 28.87486358288133 |

| Name | $\theta$ (degree) |
|------|-------------------|
| Poc1_P8_1_1_1_1 | 54.73561031724534 |
| pa_P10_3_1_1_1 | 37.377368140649665 |
| pa_P15_5_2_2_2 | 37.37736814064978 |
| Poc1_P16_3_3_3_3 | 54.73561031724534 |

# 4. Results

As we have identified nearly 1000 near-miss p-cages, it is not possible to describe them all in the main text, but a full list is provided in the electronic supplementary material. The numbers of cages found for each polygon are listed in table 2. Some of the p-cages are regular and these can be determined using basic trigonometry.

## (a) Regular connectivity-invariant homogeneous p-cages

One can first build regular p-cages from the prism hole-polyhedra. This amounts to making two pyramid-like structures, with holes, where the bases are glued together and removed. These regular p-cages are listed in table 3. All the faces are arranged symmetrically around the prism rotation axis. One can also obtain p-cages from non-symmetric arrangements of the faces, but they are all degenerate cages where some of the holes are pinched such that two opposite edges merge with each other; the resulting p-cages are equivalent to other p-cages (for example, tp_P8_1_1_3 is equivalent to Poc1_P8_1_1_1_1). A similar construction can be done with antiprism hole-polyhedra as shown at the bottom of table 3.

Placing a regular polygon on a vertex of a Platonic solid is equivalent to placing it on the faces of its dual polyhedron. This is quite an easy problem to solve and the results are provided on the left-hand side of table 4.

**Table 4.** Left: Regular connectivity-invariant p-cages derived from Platonic solids (except the `Pcu` ones), $P = 6$–17. Centre: Regular p-cages obtained from the truncated Platonic hole-polyhedra $P = 6$–17. $\theta$ is the angle between the p-cage faces and the face of the underlying solid. Right: Regular p-cages obtained from cuboctahedron, rhombicuboctahedron and rhombicosidodecahedron hole-polyhedra. Not all values of $P$ lead to regular p-cages. ($P = 18$–20 is provided in the electronic supplementary material.)

| Name |
| --- |
| Pte_P6_1_1_1 |
| Pdo_P6_1_1_1 |
| Poc1_P8_1_1_1_1 |
| Pte_P9_2_2_2 |
| Pdo_P9_2_2_2 |
| Pic_P10_1_1_1_1_1 |
| Pte_P12_3_3_3 |
| Poc1_P12_2_2_2_2 |
| Pdo_P12_3_3_3 |
| Pte_P15_4_4_4 |
| Pdo_P15_4_4_4 |
| Pic_P15_2_2_2_2_2 |
| Poc1_P16_3_3_3_3 |

| Name | $\theta$ | Dihedral angle |
| --- | --- | --- |
| Att_P8_1_2_2 | 54.736° | 0.000° |
| Ato_P8_1_2_2 | 0.000° | 90.000° |
| Att_P10_1_3_3 | 37.377° | 34.716° |
| Ato_P11_2_3_3 | 29.945° | 30.110° |
| Att_P12_1_4_4 | 0.000° | 109.471° |
| Atc_P12_1_4_4 | 0.000° | 70.529° |
| Atd_P12_1_4_4 | 0.000° | 41.810° |
| Att_P13_2_4_4 | 49.331° | 10.810° |
| Ato_P14_3_4_4 | 37.110° | 15.779° |
| Att_P15_2_5_5 | 37.377° | 34.716° |
| Att_P16_3_5_5 | 54.736° | 0.000° |
| Ato_P16_3_5_5 | 0.000° | 90.000° |
| Att_P17_2_6_6 | 21.180° | 67.112° |
| Ato_P17_4_5_5 | 40.960° | 8.080° |
| Atc_P17_2_6_6 | 21.180° | 28.170° |

| Name |
| --- |
| Arco_P8_1_1_1_1 |
| Aco_P12_1_2_3_2 |
| Arco_P12_1_2_3_2 |
| Arcd_P12_1_2_3_2 |
| Arco_P12_2_2_2_2 |
| Arco_P16_3_3_3_3 |

Placing faces on the vertices of a truncated Platonic solid is equivalent to placing a pyramid without its base on the face of the dual of the corresponding Platonic solid. The resulting p-cages are listed in the centre of table 4. One is then left with placing regular polygons on the vertices of the cuboctahedron, rhombicuboctahedron and rhombicosidodecahedron. The resulting regular p-cages are listed on the right-hand side of table 4.

The detailed geometric derivations are provided in the electronic supplementary material.

## (b) Near-miss connectivity-invariant homogeneous p-cages

We will now describe the different properties that the near-miss p-cages exhibit.

The level of deformation varies greatly between p-cages; not surprisingly, polygons with a large number of edges form more p-cages below the set deformation threshold. The number of near-miss p-cages with deformation below 1% is relatively small and we have listed them in table 5.

From the onset of our construction, we have avoided imposing that the p-cage faces are identical. We should hence find out if this was indeed justified or if the obtained p-cages do happen to have identical faces. As our computer program outputs for each p-cage the length $L_{f,k}$ of all the edges $k$ of face $f$ as well as the angles $\alpha_{f,k}$, it was easy to compare the edge length of any pair of faces $f$ and $g$ by computing $L_{f,k} - L_{g,(k+\delta)\%P}$, varying $\delta = 1 \ldots P - 1$ to determine the smallest value of the difference. We did the same with the corresponding angles $\alpha_{f,k}$ (which we now label with both an edge and a face index). The relative deformation of the p-cage faces was hence obtained by computing

$$
\left.
\begin{aligned}
\Delta L_{f,g} &= \frac{1}{P} \min_{\delta \in [1\ldots P-1]} \sum_{k=0}^{P-1} |L_{f,k} - L_{g,(k+\delta)\%P}|, \\
\Omega_l &= \max_{f,g,f \neq g} \Delta L_{f,g}, \\
\Delta \Phi_{f,g} &= \frac{1}{P\pi(1 - 2/P)} \min_{\delta \in [1..P-1]} \sum_{k=0}^{P-1} |\alpha_{f,k} - \alpha_{g,(k+\delta)\%P}|
\end{aligned}
\right\} \quad (4.1)
$$

and

$$
\Omega_a = \max_{f,g,f \neq g} \Delta \Phi_{f,g}.
$$

We found that for most cages the relative deformations were smaller than the deformations of the faces themselves, as expected, but not small, justifying our decision to impose face

**Table 5.** Near-miss p-cages with up to 1% deformation grouped by type of polygonal faces. (None for $P = 6, 7, 8$.)

| p-cage | $\Delta_l$ | $\Delta_a$ | p-cage | $\Delta_l$ | $\Delta_a$ |
|---|---|---|---|---|---|
| 7p_P9_3_1_2 | 0.0028 | 0.0028 | Arco_P15_2_3_3_3 | 0.0043 | 0.00435 |
| tp_P9_1_2_3 | 0.00863 | 0.00863 | 10p_P15_6_2_4 | 0.00511 | 0.00541 |
| Aco_P10_1_2_1_2 | $1 \times 10^{-5}$ | 0.0051 | pp_P15_5_3_4 | 0.0064 | 0.00641 |
| Atc_P10_1_3_3 | 0.00513 | 0.00513 | Ato_P15_3_2_7 | 0.00629 | 0.00642 |
| Arco_P11_1_2_2_2 | 0.00021 | 0.00109 | Att_P15_2_3_7 | 0.00721 | 0.0072 |
| Ato_P11_2_2_4 | 0.00372 | 0.00372 | Ati_P15_4_4_4 | 0.00864 | 0.00864 |
| Asc_P11_2_1_1_1_1 | 0.00496 | 0.00272 | Att_P16_2_5_6 | 0.00055 | 0.00055 |
| Att_P11_1_3_4 | 0.00508 | 0.00508 | Ati_P16_4_2_7 | 0.00074 | 0.00087 |
| Arco_P11_1_1_2_3 | 0.0052 | 0.00271 | Att_P16_2_4_7 | 0.00089 | 0.00093 |
| Ato_P11_2_1_5 | 0.00695 | 0.00249 | Ati_P16_4_3_6 | 0.00194 | 0.00194 |
| Atc_P11_1_2_5 | 0.00702 | 0.00702 | 7p_P16_6_3_4 | 0.00223 | 0.00217 |
| 8a_P11_4_1_1_1 | 0.008 | 0.00799 | Pcu_P16_3_5_5 | 0.00267 | 0.00266 |
| 8p_P11_4_1_3 | 0.00857 | 0.00383 | Ati_P16_4_4_5 | 0.00275 | 0.00275 |
| Atc_P11_1_3_4 | 0.0095 | 0.00947 | Arcd_P16_2_3_4_3 | 0.0061 | 0.00618 |
| 7p_P11_4_1_3 | 0.00978 | 0.00978 | Aco_P16_2_4_2_4 | $1 \times 10^{-5}$ | 0.00641 |
| Atd_P12_1_3_5 | 0.00358 | 0.00359 | Aco_P16_2_3_2_5 | 0.00754 | 0.00754 |
| hp_P12_4_2_3 | 0.00401 | 0.00401 | Pcu_P16_4_4_5 | 0.00802 | 0.00801 |
| Pcu_P12_3_2_4 | 0.00813 | 0.00813 | Aid_P16_2_4_2_4 | $1 \times 10^{-5}$ | 0.0086 |
| tp_P12_2_3_4 | 0.00951 | 0.00951 | hp_P16_6_2_5 | 0.00895 | 0.00895 |
| Ati_P13_3_3_4 | 0.00188 | 0.0019 | tp_P16_3_4_6 | 0.00963 | 0.00964 |
| 9p_P13_5_2_3 | 0.00217 | 0.00217 | Atc_P16_2_4_7 | 0.0098 | 0.00981 |
| Ati_P13_3_2_5 | 0.00293 | 0.00303 | Aco_P17_2_4_3_4 | $5 \times 10^{-5}$ | 0.00084 |
| tp_P13_2_3_5 | 0.00446 | 0.00446 | 10p_P17_7_3_4 | 0.00106 | 0.00103 |
| 8p_P13_5_2_3 | 0.00775 | 0.00775 | tp_P17_3_5_6 | 0.0012 | 0.0012 |
| 9p_P13_4_3_3 | 0.00941 | 0.00941 | Atc_P17_2_5_7 | 0.00133 | 0.00136 |
| 10p_P13_5_2_3 | 0.00883 | 0.00943 | Atc_P17_2_4_8 | 0.00129 | 0.00152 |
| pp_P13_4_2_4 | 0.00955 | 0.00955 | tp_P17_3_4_7 | 0.00247 | 0.00247 |
| hp_P14_5_2_4 | 0.00398 | 0.00397 | 9p_P17_7_2_5 | 0.00342 | 0.00338 |
| tp_P14_2_4_5 | 0.00688 | 0.00688 | 10a_P17_7_2_2_2 | 0.00367 | 0.0037 |
| sa_P14_4_2_2_2 | 0.0073 | 0.0073 | 10p_P17_7_2_5 | 0.00451 | 0.00322 |
| Pcu_P14_4_2_5 | 0.00482 | 0.00904 | 9a_P17_7_2_2_2 | 0.0045 | 0.00459 |
| Ato_P15_3_4_5 | 0.0011 | 0.0011 | Pcu_P17_5_4_5 | 0.00665 | 0.00665 |
| 9p_P15_6_2_4 | 0.00227 | 0.00227 | 9p_P17_7_3_4 | 0.00676 | 0.00674 |
| hp_P15_4_4_4 | 0.00263 | 0.00262 | Ati_P17_4_5_5 | 0.00686 | 0.00686 |
| Ato_P15_3_3_6 | 0.00272 | 0.00271 | Pcu_P17_5_3_6 | 0.00761 | 0.0076 |
| Aco_P15_2_3_2_4 | 0.00281 | 0.00282 | Att_P17_2_5_7 | 0.00782 | 0.00786 |
| Pcu_P15_4_3_5 | 0.00419 | 0.00419 | hp_P17_6_3_5 | 0.00932 | 0.00932 |

connectivity invariance at graph level rather than geometrically. The only p-cages for which $\Omega_l$ and $\Omega_a$ are very small are some of the ones corresponding to a tiling of the face of a Platonic solid and some for which the angles are all regular. Full details are provided in the electronic supplementary material, which describes all the near-miss p-cages. To test the accuracy of our minimization, we have performed it several times on the same p-cages and obtained the same result each time. Moreover, to evaluate the numerical accuracy of our procedure, we have also relaxed the regular p-cages and obtained deformations $\Delta_l$ or $\Delta_a$ equal to $10^{-6}$ or smaller.

The p-cages with the smallest deformation for each value of $P$ are presented in figure 10. We can see from table 5 and figure 10 that the deformations are very small, below 0.1% for several of them, and are so small that they are impossible to detect with the naked eye. We also see that the p-cages exhibit a variety of features, which we will now describe.

In figure 11*a*, we present some p-cages obtained from prism and antiprism hole-polyhedra. They all appear as rings, which is well illustrated by 9p_P12_5_1_3 and 9p_P15_6_2_4. These rings can then be nearly flat, like 7p_P9_2_2_2, or elongated, like tp_P15_5_3_4. P-cages obtained from antiprisms look similar except that they have four neighbours, forcing them to assume ring-like structures such as 8a_P11_4_1_1_1.

Several p-cages derived from prisms look very similar to p-cages obtained from antiprisms, the only difference being that the latter p-cages have an extra set of joined faces which for the former p-cage becomes a tiny gap. They are listed on the left-hand side of table 6.

Figure 11*b* presents some typical p-cages derived from Platonic solid hole-polyhedra. They all correspond to an embedding of the polygon into the faces of the dual of the hole-polyhedron.

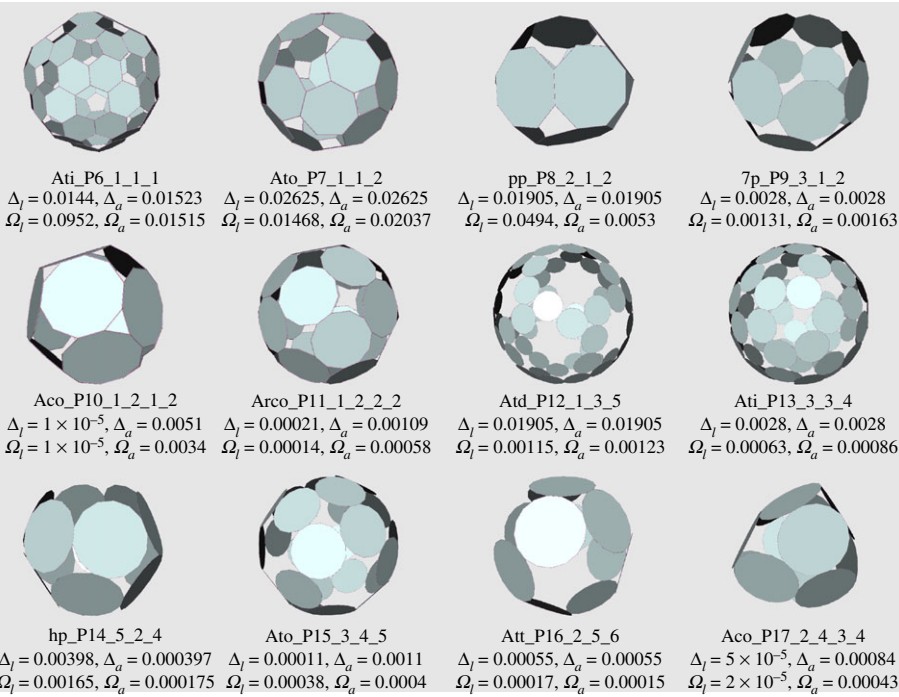

**Figure 10.** Least deformed (smallest $\Delta_l + \Delta_a$) near-miss p-cages for each value of $P$. While they do look regular they are not so. (Online version in colour.)

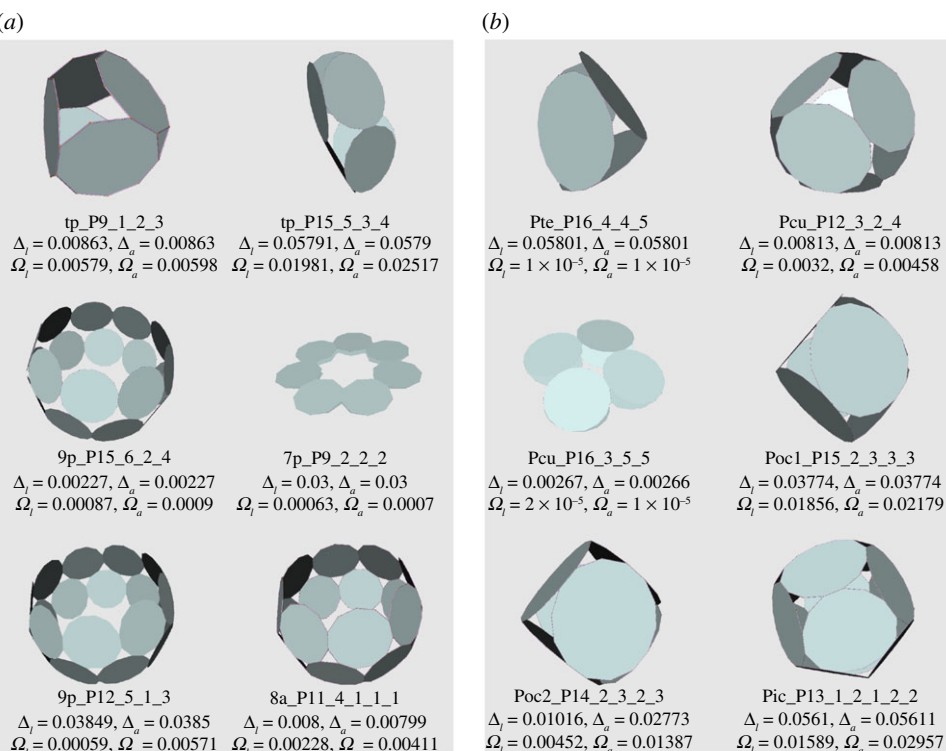

**Figure 11.** Some p-cages obtained from: (*a*) prisms and antiprisms, (*b*) Platonic solids. (Online version in colour.)

**Table 6.** Visual similarity between p-cages with different numbers of neighbour faces. Left: Prism-derived p-cages and antiprism-derived p-cages. Right: Archimedean solid-derived p-cages. Parenthesis denotes deformations exceeding 10%.

| | | | |
|---|---|---|---|
| 7p_P10_3_1_3 | 7a_P10_3_1_1_1 | Arco_P11_1_1_2_3 | Asc_P11_2_1_1_1_1 |
| 8p_P10_3_1_3 | 8a_P10_3_1_1_1 | Ato_P11_2_1_5 | Asc_P11_2_1_1_1_1 |
| pp_P11_4_1_3 | pa_P11_4_1_1_1 | Ati_P11_2_1_5 | Asd_P11_2_1_1_1_1 |
| hp_P11_4_1_3 | ha_P11_4_1_1_1 | Arcd_P11_1_1_2_3 | Asd_P11_2_1_1_1_1 |
| 7p_P11_4_1_3 | 7a_P11_4_1_1_1 | Ati_P12_3_1_5 | Asd_P12_3_1_1_1_1 |
| 8p_P11_4_1_3 | 8a_P11_4_1_1_1 | Arcd_P12_1_1_3_3 | Asd_P12_3_1_1_1_1 |
| 9p_P11_4_1_3 | 9a_P11_4_1_1_1 | Arco_P12_1_1_3_3 | Asc_P12_3_1_1_1_1 |
| 10p_P11_4_1_3 | 10a_P11_4_1_1_1 | Arco_P11_2_1_3_3 | Asc_P13_3_1_2_1_1 |
| hp_P12_5_1_3 | ha_P12_5_1_1_1 | Arcd_P13_2_1_3_3 | Asd_P13_3_1_2_1_1 |
| 8p_P12_5_1_3 | 8a_P12_5_1_1_1 | Arcd_P13_1_1_4_3 | (Asd_P13_4_1_1_1_1) |
| 7p_P12_5_1_3 | 7a_P12_5_1_1_1 | Ati_P14_4_1_6 | Asd_P14_4_1_2_1_1 |
| 9p_P12_5_1_3 | 9a_P12_5_1_1_1 | Arcd_P14_2_1_4_1 | Asd_P14_4_1_2_1_1 |
| 10p_P12_5_1_3 | 10a_P12_5_1_1_1 | Ato_P14_4_1_6 | Asc_P14_4_1_2_1_1 |
| 8p_P13_6_1_3 | 8a_P13_6_1_1_1 | Arco_P14_3_1_3_3 | Asc_P14_3_1_3_1_1 |
| 9p_P13_6_1_3 | 9a_P13_6_1_1_1 | Arcd_P14_2_2_2_4 | (Asd_P14_2_2_2_1_2) |
| 10p_P13_6_1_3 | 10a_P13_6_1_1_1 | Arcd_P15_1_2_3_5 | Asd_P15_3_2_1_2_2 |
| hp_P15_5_2_5 | ha_P15_5_2_2_2 | Arco_P15_1_2_3_5 | Asc_P15_2_2_2_2_2 |
| 7p_P15_5_2_5 | 7a_P15_5_2_2_2 | Arcd_P15_3_1_4_3 | Asd_P15_4_1_3_1_1 |
| 8p_P15_5_2_5 | 8a_P15_5_2_2_2 | Arco_P15_3_1_4_3 | Asc_P15_4_1_3_1_1 |
| pp_P16_6_2_5 | pa_P16_6_2_2_2 | Ati_P15_5_1_6 | Asd_P15_5_1_2_1_1 |
| 7p_P16_6_2_5 | 7a_P16_6_2_2_2 | Arcd_P15_2_1_5_3 | Asd_P15_5_1_2_1_1 |
| 8p_P16_6_2_5 | 8a_P16_6_2_2_2 | Arcd_P16_2_2_4_4 | Asd_P16_4_1_3_1_1 |
| 9p_P16_6_2_5 | 9a_P16_6_2_2_2 | Arco_P16_2_2_3_5 | Asc_P16_3_2_2_2_2 |
| 10p_P16_6_2_5 | 10a_P16_6_2_2_2 | Arcd_P16_2_2_3_5 | Asd_P16_3_2_2_2_2 |
| 7p_P17_7_2_5 | 7a_P17_7_2_2_2 | Arcd_P16_1_2_4_5 | Asd_P16_4_2_1_2_2 |
| 8p_P17_7_2_5 | 8a_P17_7_2_2_2 | Arcd_P16_3_1_5_3 | Asd_P16_5_1_3_1_1 |
| 9p_P17_7_2_5 | 9a_P17_7_2_2_2 | Ato_P17_4_2_8 | Asc_P17_4_2_2_2_2 |
| 10p_P17_7_2_5 | 10a_P17_7_2_2_2 | Arcd_P17_2_2_4_5 | Asd_P17_4_2_2_2_2 |
| | | Ati_P17_4_2_8 | Asd_P17_4_2_2_2_2 |
| | | Arco_P17_2_2_4_5 | Asc_P17_4_2_2_2_2 |
| | | Arco_P17_1_2_5_5 | Asc_P17_5_2_1_2_2 |

When the numbers of hole edges are all equal, the p-cage is regular. P16_3_5_5 is different because the cube is also a square prism and this allows it to flatten like other prism-based p-cages, but this cannot happen for the other Platonic-based p-cages.

Figures 12 and 13 present a range of p-cages derived from Archimedean solid hole-polyhedra. The majority of p-cages assume a sphere-like shape, such as Atc_P17_2_4_8, Ato_P17_5_2_7 or Asc_P11_2_1_1_1_1, but for some specific hole edge distributions the p-cage can look like a tiling of the faces of a Platonic solid, such as Att_P14_1_5_5 and Ato_P13_2_4_4 (figure 12), where each edge of the Platonic solid is where two faces of the p-cage are joined together, or Ati_P11_2_3_3, Aco_P15_1_3_4_3, Arco_P14_3_2_3_2 and Arcd_P17_2_3_5_3 (figure 13), where each edge of the Platonic solid is where two pairs of faces of the p-cage are joined together. Some others, on the other hand, look like a wire-frame construction of the Archimedean solids, such as Att_P11_2_3_3, Ato_P9_2_2_2 or Atd_P10_1_3_3.

As we can see from all the figures, we also note that some cages have very small holes while others have very large ones. For some cages with holes with a large *Q* value, the faces organize themselves to fill the gap of what could potentially be a very large hole. This is the case for Ati_P17_5_2_7, where $Q = 6$ for one group of holes, but where each face seems to have five neighbours when they actually have only three.

## 5. Conclusion

In this paper, we have defined near-miss connectivity-invariant p-cages as assemblies of nearly regular polygons with holes between them where all the faces are connectivity equivalent. We have then shown that each p-cage can be characterized by a planar graph, the hole-polyhedron, where the holes' edges are distributed around the nodes of the graph. We have

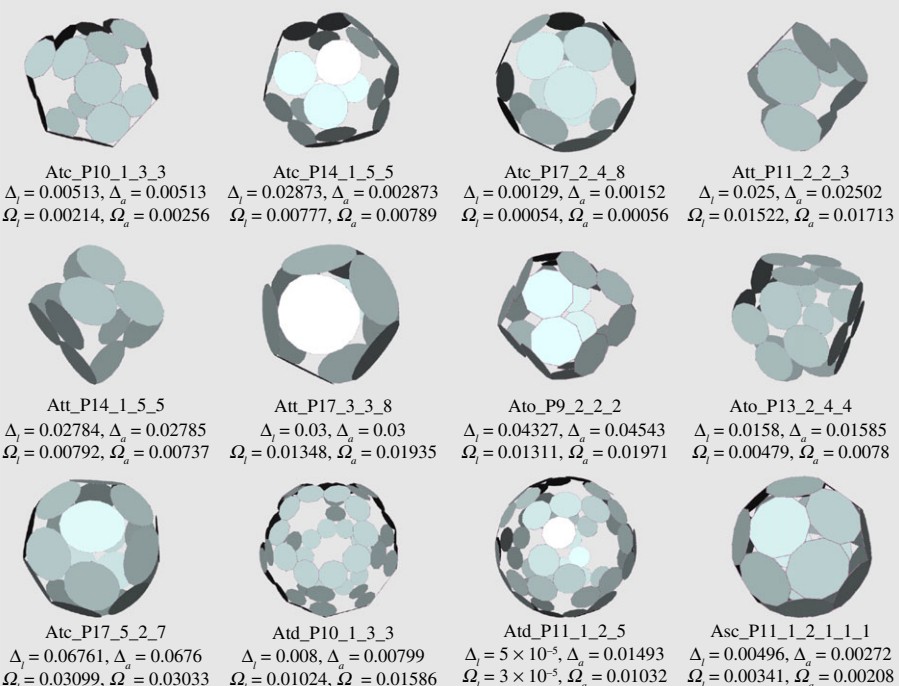

**Figure 12.** Some p-cages obtained from Archimedean solids (part I). (Online version in colour.)

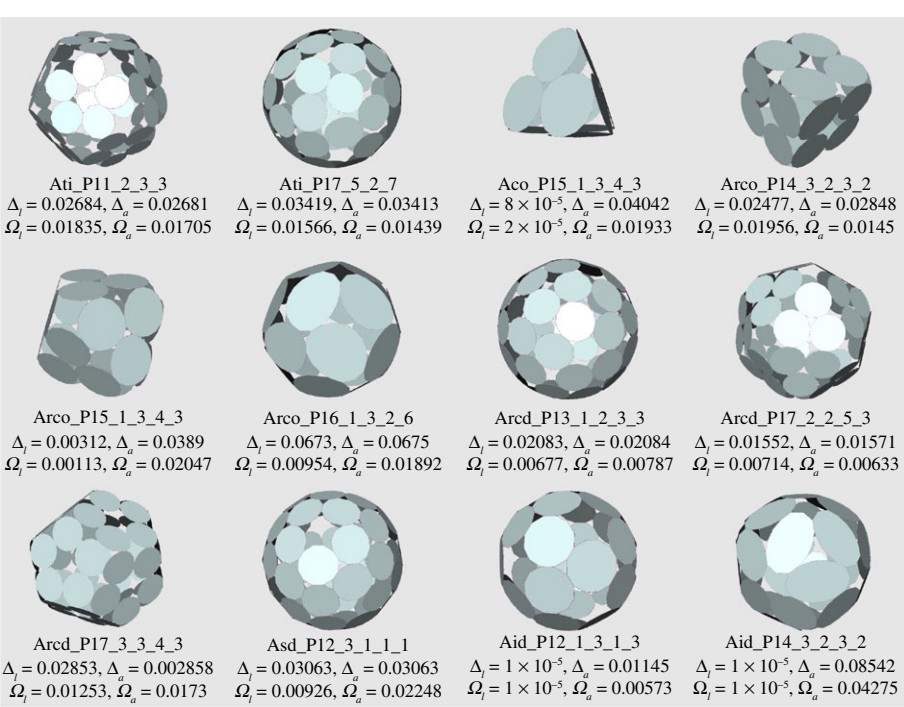

**Figure 13.** Some p-cages obtained from Archimedean solids (part II). (Online version in colour.)

then enumerated all the distributions of hole edges on the hole-polyhedra compatible with the connectivity invariance of the p-cages, excluding those which would necessarily lead to edge length and angle deformation exceeding 10% and restricting ourselves to the polygons with 6–17 edges.

We have then derived a quality function which measures the level of deformation of the p-cages and have used a numerical method to minimize that quality function for each of the possible configurations we had identified. This resulted in a large number of non-regular p-cages, most of which had a deformation exceeding a 10% threshold that we had set upfront, but still leaving around 1000 near-miss p-cages with deformation below 10% and 74 near-miss p-cages with deformation less than 1%.

We proceeded by describing some properties of the obtained p-cages with 6–17 edges. Most near-miss p-cages have configurations similar to the regular p-cages, but some are different in that large holes are filled with the faces, leaving what looks like medieval castle loopholes.

In our approach, we have not assumed any symmetry for the deformed cages, as the different faces of a p-cage could potentially be deformed differently. We have found that, for most cages, the faces were deformed slightly differently and that our assumption was thus justified.

We have thus generated a very large list of potential geometries for nearly symmetric protein cages. While some p-cages exhibit large holes, probably making them of lesser use in biochemistry, many others have a pseudo-spherical shape, making them good geometrical candidates for shells which could contain some cargo.

Data accessibility. All scripts used in this study are openly accessible through https://github.com/StochasticBiology/boolean-efflux.git. The data are provided in the electronic supplementary material [17]. The C++ and Python programs used to generate all the data are available from doi:10.6084/m9.figshare.14061782.

Authors' contributions. B.M.A.G.P.: conceptualization, formal analysis, investigation, methodology, project administration, resources, software, supervision, validation, visualization, writing—original draft, writing—review and editing; A.K.: formal analysis, investigation, software, visualization, writing—original draft, writing—review and editing; J.G.H.: conceptualization, funding acquisition, project administration, writing—original draft, writing—review and editing.

All authors gave final approval for publication and agreed to be held accountable for the work performed herein.

Conflict of interest declaration. The authors declare the following competing financial interests: J.G.H. is named as an inventor on a number of patent applications related to protein-cage assembly, decoration and filling. J.G.H. is also the founder of and holds equity in nCage Therapeutics LLC, which aims to commercialise protein cages for therapeutic applications.

Funding. J.G.H. and A.K. were funded by National Science Centre (NCN, Poland) grant no. 2016/20/W/NZ1/00095 (Symfonia-4) awarded to J.G.H.

Acknowledgements. We want to thank Symfonia team members from the Bionanoscience and Biochemistry Laboratory (BBL). B.M.A.G.P. would like to thank the BBL for its hospitality. The computer simulations were performed on the Condor cluster of the Mathematical Science Department of Durham University. The figures were produced with the Antiprism Polyhedron Modelling Software (www.antiprism.com/).

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
