## [Peer Review File · Proceedings. Mathematical, Physical, and Engineering Sciences]

Review History

RSPA-2021-0156.R0 (Original submission)

Review form: Referee 1

Is the manuscript an original and important contribution to its field?

Marginal

Is the paper of sufficient general interest?

Good

Is the overall quality of the paper suitable?

Acceptable

Can the paper be shortened without overall detriment to the main message?

Yes

Do you think some of the material would be more appropriate as an electronic appendix?

No

Do you have any ethical concerns with this paper?

No

Recommendation?

Major revision is needed (please make suggestions in comments)

Comments to the Author(s)

Slightly irregular (or 'near-miss') invariant p-cages are of considerable interest in molecular biology; even compared to regular ones, since irregular cages can produce a more evenly distributed covering by P-sided polygons for some values of P (a really nice 'near-miss' is the case P=11 with 24 faces that the motivation of the research comes from).

#1 General comments

The main idea is the sufficient distortion of hole edges to let shared edges of p-cages be tuned according to the conditions of continuity, planarity and convexity is elegant. However, there are still major concerns about the applied optimization method. It is not clear what kind of connectivity is assumed for the initial configurations. Final result must sensibly depend on topological conditions. The optimization procedure looks rather heuristic: why $c_l + c_a$ is adopted specifically? Relative equivalence of perturbations of leg lengths and right angles in an isosceles right triangle is credibly shown in the paper but angular deviation of adjacent edges of, e.g., a 17-gon is $2\pi/17 < \pi/2$. It still might work but qualitatively is not convincing. 'Quality functions', even if they have some, are not related to mechanical properties as an energy functional, so it is impossible to judge what combinations of weights are reasonable. (The convergence of the method is something but it rises some discomfort when exact solutions supported by symmetry are only found by a remarkable threshold, see the example p-cage mentioned in the first paragraph.)

As a summary of all above, a 'nearly fatal' misconception is explicitly declared at the end of the paper (page 15, lines 35-37). Some of the near-miss cages are PROVEN to have symmetry (see [1]) but the iterative method leaves this property unrevealed. Although there is no evidence to happen so always but it seems to be a strong conjecture that MOST p-cages shown here has the same level of metric symmetry as the symmetry of its connectivity graph is. It is even strengthened by the aim of dealing with 'invariant' cages only (it is true, however, that invariance is not a direct proof of a metric vertex-transitivity of near-miss cages). It can also be concluded that a much more efficient iteration could have been programmed based on a few geometric parameters using symmetry arguments.

A possible idea is as follows: look at the name code of each near-miss p-cage and depart from the symmetry group of order n it belongs to. Any plane H can be given in 3D by 3 scalars (this number can perhaps be reduced by considerations in elementary geometry, especially for regular p-cages). In the following step, based on all n images of H under symmetry operations, lines of intersection and even the midpoint of shared edge can be found (analytically?) between connected faces (if no shared edge is assumed, no intersection is considered). If the hole polyhedron is L_h -regular, intersections define a set of n copies of L_h -gons. In the last step, 'inscribed' irregular P-gons are fitted to L_h -gons (respecting L_h edge midpoints as well) for different values of P. In this way, symmetry and planarity will never be violated, and the method possibly offers an iteration in much less variables.

#2 Technical remarks

The notation of the ms is positively sloppy: it starts, e.g., by denoting the vertex number of polygons by P then converts to N gradually in Section 3 without giving any notice. Symbols for cyclic subscripts are inconsistent, formulae frequently miss some letters or use some of them

incorrectly. Some details and suggestions are given in the following list.

P4 Fig. 1b It is better to have to same number of edges as in 1a, i.e., 5. The exact meaning of ϵ_i cannot be seen in part b.

P4 147: LHS of (2.1) should be duplicated

P4 151: It would be better to introduce q before (2.1)

P5 111: The equation is OK only if β_{i0} (and not β_i) is $2\pi/P$. It should be stated explicitly what do α_{i0} , β_{i0} , γ_{i0} stand for. Note that if ' i_0 ' refers to the original (regular) configuration, then κ_i and $\kappa_{\alpha,i}$ must have opposite sign, since α and γ cannot increase at the same time.

P5 (2.6) π instead of β_i (or 2 instead of P , cf. (2.3))

P5 (2.7) Missing factor 2 on the RHS

P5 131: Isn't it 'sum of ϵ other edge lengths' of the same sub-faces hole'?

P5 135: Equality can only hold if all γ s are 0 (provided A is understood as shown in Fig. 1c)

P5 140-41: why not ' A_j ' on the left hand side (A_i is on the RHS)?

P5 149: 'majorate or minorate': isn't it governed by the sign of κ_{i0} ??

P5 (2.10) is true only if shared edges of P -gons are of unit length (but L_f is only defined later on P7 131). Moreover, (2.10) and (2.11) should be merged by summing terms from $i=0$ to q_k-1 .

Generally speaking, 'deformations' are first mentioned in this section on P5, 110. It would greatly help understanding if a clear statement was made here what kind of deformations are considered (it seems that unit edge length of P -gons is kept and A_i is changed but it is not written explicitly). A figure should be attached as well to show the meaning of A_i , angles with subscript ' i ' and ' i_0 ', all magnification factors κ , etc.

P6 135: Isn't strict planarity required? It is not subject then to minimization...

P6 138: The definition of N_N is a bit unclear: do nodes count twice if they are on a shared edge? (3.2) could make it clear if the summation was expanded (however, this is not a relevant question as far as the cage is 'sufficiently' symmetric). Similarly, in (3.3) and (3.7), $\sum_{i=1}^P$ (or $\sum_{i=0}^{P-1}$) is better instead of 'face nodes'.

P6-7: There is a mismatch between (3.1) and (3.6): $(i+1)/N_f \ll i/N_f+1$ (and their ranges $0..N_f-1$ and $1..N_f$ are also different). N_f could be P_f according to the initial notation.

P7 126: 'optimize the symmetry of the cage'??

P7 143: 'isosceles rectangular triangle'  '...right triangle'?

P7 156 'in radians'

P8 (3.14) seems to penalize convexity, not concavity

P8 131: Is $d = M-F$? Please give a definition. If the present condition does not strictly impose convexity, why not to consider $H((W_i \times W_i) \cdot s)$?

P9 17: (d)  (3.18); 'where'  'were'

P9, 114 as well as 141: 'edge lengths'

P9 (3.19) is only necessary to get separate limits (both set later on to 0.1) for Δ_l and Δ_a . It would have been simpler and more straightforward to constrain $(\Delta_l^2 + \Delta_a^2)$, directly following from the functional instead of $|\Delta_l| + |\Delta_a|$.

P9 (3.20) uses a misleading (although not incorrect) notation: i stood earlier on for individual edges of a face f , A_i denoted a diagonal ('edge of sub-faces hole'), not an angle: this later was $\alpha_{f,i}$ as defined in (3.1). $ij \rightarrow fg$, as well as $k \rightarrow i$ would be more consequent. P and N within the same formula??!

The text preceding (3.20) would be easier to follow if its motivation was given: since those p -cages may exhibit some topological symmetry, they are expected to retain some metric symmetry as well. In order to quantify the differences properly, a preceding identification of 'nearly similar' faces should be performed: one face is unchanged, another is successively rotated until a 'quasi-overlapping' is detected. It should explicitly be written in the text.

P9 157: 'tiling'

P10, Figure 2: all p-cages listed here exhibit symmetries of some Archimedean solids (in a general sense, including those of prisms, antiprisms and even of Platonic solids as well): how can some of them have notable differences between faces (Ω_a and Ω_l both exceed 1% in $A_{P7_1_1_2}$)? What kind of terminal condition for the iteration was used?

P12 l50: $A_{2_4_4}$ must be $A_{P13_2_4_4}$

P13 l58: 'all the possible cage connectivities': it is not proved anywhere.

Decision letter (RSPA-2021-0156.R0)

25-May-2021

Dear Professor Piette:

I am writing to inform you that your manuscript RSPA-2021-0156 entitled "Characterisation of Invariant Homogeneous Convex Polyhedral Cages. II: Near-miss Cages" has been rejected in its present form for publication in Proceedings A.

The Editor has made this decision based on the advice of referees, and taking into account their own opinion of your paper. With this in mind we would like to invite a resubmission, provided the comments of the referees and any comments from the Editor are taken into account. This is not a provisional acceptance.

The resubmission will be treated as a new manuscript. Please note that resubmissions must be submitted within six months of the date of this email. In exceptional circumstances, extensions may be possible if agreed with the Editorial Office.

Please find below the comments made by the referees, not including confidential reports to the Editor, which I hope you will find useful. If you do choose to resubmit your manuscript, please include details of how you have responded to the comments, and the adjustments you have made.

Please note that we have a strict upper limit of 28 pages for each paper. Please endeavour to incorporate any revisions while keeping the paper within journal limits. Please note that page charges are made on all papers longer than 20 pages. If you cannot pay these charges you must reduce your paper to 20 pages before submitting your revision. Your paper has been ESTIMATED to be 16 pages. We cannot proceed with typesetting your paper without your agreement to meet page charges in full should the paper exceed 20 pages when typeset. If you have any questions, please do get in touch.

To upload a resubmitted manuscript, log into <http://mc.manuscriptcentral.com/prsa> and enter your Author Centre, where you will find your manuscript title listed under "Manuscripts with Decisions." Under "Actions," click on "Create a Resubmission." Please be sure to indicate that it is a resubmission, and ensure you enter this ID - RSPA-2021-0156 - as the previous submission number.

Yours sincerely
Raminder Shergill
proceedingsa@royalsociety.org

Reviewer(s)' Comments to Author:

Referee: 1

Comments to the Author(s)

Slightly irregular (or 'near-miss') invariant p-cages are of considerable interest in molecular biology; even compared to regular ones, since irregular cages can produce a more evenly distributed covering by P-sided polygons for some values of P (a really nice 'near-miss' is the case P=11 with 24 faces that the motivation of the research comes from).

#1 General comments

The main idea is the sufficient distortion of hole edges to let shared edges of p-cages be tuned according to the conditions of continuity, planarity and convexity is elegant.

However, there are still major concerns about the applied optimization method.

It is not clear what kind of connectivity is assumed for the initial configurations. Final result must sensibly depend on topological conditions.

The optimization procedure looks rather heuristic: why $c_l + c_a$ is adopted specifically? Relative equivalence of perturbations of leg lengths and right angles in an isosceles right triangle is credibly shown in the paper but angular deviation of adjacent edges of, e.g., a 17-gon is $2\pi/17$;

As a summary of all above, a 'nearly fatal' misconception is explicitly declared at the end of the paper (page 15, lines 35-37). Some of the near-miss cages are PROVEN to have symmetry (see [1]) but the iterative method leaves this property unrevealed. Although there is no evidence to happen so always but it seems to be a strong conjecture that MOST p-cages shown here has the same level of metric symmetry as the symmetry of its connectivity graph is. It is even strengthened by the aim of dealing with 'invariant' cages only (it is true, however, that invariance is not a direct proof of a metric vertex-transitivity of near-miss cages). It can also be concluded that a much more efficient iteration could have been programmed based on a few geometric parameters using symmetry arguments.

A possible idea is as follows: look at the name code of each near-miss p-cage and depart from the symmetry group of order n it belongs to. Any plane H can be given in 3D by 3 scalars (this number can perhaps be reduced by considerations in elementary geometry, especially for regular p-cages). In the following step, based on all n images of H under symmetry operations, lines of intersection and even the midpoint of shared edge can be found (analytically?) between connected faces (if no shared edge is assumed, no intersection is considered). If the hole polyhedron is L_h -regular, intersections define a set of n copies of L_h -gons. In the last step, 'inscribed' irregular P-gons are fitted to L_h -gons (respecting L_h edge midpoints as well) for different values of P. In this way, symmetry and planarity will never be violated, and the method possibly offers an iteration in much less variables.

#2 Technical remarks

The notation of the ms is positively sloppy: it starts, e.g., by denoting the vertex number of polygons by P then converts to N gradually in Section 3 without giving any notice. Symbols for cyclic subscripts are inconsistent, formulae frequently miss some letters or use some of them incorrectly. Some details and suggestions are given in the following list.

P4 Fig. 1b It is better to have to same number of edges as in 1a, i.e., 5. The exact meaning of ϵ_i cannot be seen in part b.

P4 147: LHS of (2.1) should be duplicated

P4 151: It would be better to introduce q before (2.1)

P5 111: The equation is OK only if β_{i0} (and not β_i) is $2\pi/P$. It should be stated explicitly what do α_{i0} , β_{i0} , γ_{i0} stand for. Note that if ' i_0 ' refers to the original (regular)

configuration, then κ_i and $\kappa_{\alpha,i}$ must have opposite sign, since α and γ cannot increase at the same time.

P5 (2.6) π instead of β_i (or 2 instead of P, cf. (2.3))

P5 (2.7) Missing factor 2 on the RHS

P5 l31: Isn't it 'sum of $_other$ edge lengths $_$ of the same sub-faces hole'?

P5 l35: Equality can only hold if all γ s are 0 (provided A is understood as shown in Fig. 1c)

P5 l40-41: why not ' A_j ' on the left hand side (A_i is on the RHS)?

P5 l49: 'majorate or minorate': isn't it governed by the sign of $\kappa_{\alpha,i}$??

P5 (2.10) is true only if shared edges of P-gons are of unit length (but L_f is only defined later on P7 l31). Moreover, (2.10) and (2.11) should be merged by summing terms from $i=0$ to q_k-1 .

Generally speaking, 'deformations' are first mentioned in this section on P5, l10. It would greatly help understanding if a clear statement was made here what kind of deformations are considered (it seems that unit edge length of P-gons is kept and A_i is changed but it is not written explicitly). A figure should be attached as well to show the meaning of A_i , angles with subscript 'i' and 'i0', all magnification factors κ , etc.

P6 l35: Isn't strict planarity required? It is not subject then to minimization...

P6 l38: The definition of N_N is a bit unclear: do nodes count twice if they are on a shared edge? (3.2) could make it clear if the summation was expanded (however, this is not a relevant question as far as the cage is 'sufficiently' symmetric). Similarly, in (3.3) and (3.7), $_i=1^P$ (or $_i=0^P-1$) is better instead of 'face nodes'.

P6-7: There is a mismatch between (3.1) and (3.6): $(i+1)/N_f$ <> $i\%N_f+1$ (and their ranges $0..N_f-1$ and $1..N_f$ are also different). N_f could be P_f according to the initial notation.

P7 l26: 'optimize the symmetry of the cage'??

P7 l43: 'isosceles rectangular triangle'  '...right triangle'?

P7 l56 'in radians'

P8 (3.14) seems to penalize convexity, not concavity

P8 l31: Is $d = M-F$? Please give a definition. If the present condition does not strictly impose convexity, why not to consider $H((W_i \times W_i) \setminus \text{cdot } s)$?

P9 l7: (d)  (3.18); 'where'  'were'

P9, l14 as well as l41: 'edge lengths'

P9 (3.19) is only necessary to get separate limits (both set later on to 0.1) for Δ_l and Δ_a . It would have been simpler and more straightforward to constrain $(\Delta_l^2 + \Delta_a^2)$, directly following from the functional instead of $|\Delta_l| + |\Delta_a|$.

P9 (3.20) uses a misleading (although not incorrect) notation: i stood earlier on for individual edges of a face f , A_i denoted a diagonal ('edge of sub-faces hole'), not an angle: this later was $\alpha_{f,i}$ as defined in (3.1). ij  f,g , as well as k  i would be more consequent. P and N within the same formula??!

The text preceding (3.20) would be easier to follow if its motivation was given: since those p-cages may exhibit some topological symmetry, they are expected to retain some metric symmetry as well. In order to quantify the differences properly, a preceding identification of 'nearly similar' faces should be performed: one face is unchanged, another is successively rotated until a 'quasi-overlapping' is detected. It should explicitly be written in the text.

P9 l57: 'tiling'

P10, Figure 2: all p-cages listed here exhibit symmetries of some Archimedean solids (in a general sense, including those of prisms, antiprisms and even of Platonic solids as well): how can some of them have notable differences between faces (Ω_a and Ω_l both exceed 1% in $A_{to_P7_1_1_2}$)? What kind of terminal condition for the iteration was used?

P12 l50: $A_{to_2_4_4}$ must be $A_{to_P13_2_4_4}$

P13 l58: 'all the possible cage connectivities': it is not proved anywhere.

Author's Response to Decision Letter for (RSPA-2021-0156.R0)

See Appendix A.

RSPA-2021-0679.R0

Review form: Referee 1

Is the manuscript an original and important contribution to its field?

Acceptable

Is the paper of sufficient general interest?

Acceptable

Is the overall quality of the paper suitable?

Acceptable

Can the paper be shortened without overall detriment to the main message?

Yes

Do you think some of the material would be more appropriate as an electronic appendix?

No

Do you have any ethical concerns with this paper?

No

Recommendation?

Accept with minor revision (please list in comments)

Comments to the Author(s)

The changes made by the authors before resubmission contributed to improve the manuscript: the numerical efforts and results of identifying near-miss p-cages are now better emphasized. There is a single question about some change in the data: the original manuscript mentions 5160 potential configurations for possible candidates of near-miss p-cages, which is raised to 5743 in the revised ms - where did the increment come from?

The answers given to the referee's questions are mainly acceptable; however, there are still some concerns with Section 1(c) as far as the completeness of theory of regular invariant p-cage graphs are concerned. Two objections which has to be resolved before publication:

P6 Figure 4: even if it is redrawn, ' $x=a, y=b, \dots$ ' - remains in wrong position (should pertain to triangle #2, not #3)

P5 Figure 3: if some hexagons in Archimedean solids can have 'bcbcbc', why bases of prisms cannot? Any pattern 'bcbc...bc' for $2n$ -sided prisms (completed by 'a's accordingly) would be a connectivity-invariant configuration, too (even if they are perhaps not metrically realizable). Does it make a change in the catalog of regular connectivity-invariant p-cages?

Typesetting:

P4 l36: p-cage geometries (also P10 l21: p-cage configurations)

References [17]: Möller instead of Moller

The reviewer apologizes for his earlier mistaken comment about P17, Table 11: the question should have been asked as why a P12_2_3_4 arrangement (giving a truncated icosidodecahedron as hole-polyhedron) is not part of the regular p-cages. Explicit exclusion of the above-mentioned Archimedean solid makes the question irrelevant, though.

Review form: Referee 2

Is the manuscript an original and important contribution to its field?

Excellent

Is the paper of sufficient general interest?

Excellent

Is the overall quality of the paper suitable?

Excellent

Can the paper be shortened without overall detriment to the main message?

Yes

Do you think some of the material would be more appropriate as an electronic appendix?

No

Do you have any ethical concerns with this paper?

No

Recommendation?

Accept as is

Comments to the Author(s)

This article presents the results of a numerical study of polyhedral cages whose faces are almost regular. It is motivated by the recent discovery of a protein cage with these properties. The investigation employs a combination of combinatorial and numerical methods.

To each polyhedral cage the authors associate a dual graph. If the polyhedral cage is connectivity-invariant then the associated graph will be planar and vertex-transitive, and such graphs have been classified by Maschke. The authors proceed to give a combinatorial classification of the possible homogeneous polyhedral cages corresponding to Maschke's graphs. They then use a numerical method to find almost-regular geometric polyhedral cages corresponding to the abstract polyhedral cages in their classification. Their combinatorial classification is clear and well-explained, and their numerical strategy seems well-adapted to the task. In particular, by using a metropolis algorithm they can be reasonably confident of having identified the majority of almost-regular polyhedral cages within their search parameters.

The end result is that a surprisingly large number of almost regular polyhedral cages have been found. So the experimentally-discovered protein cage is not an isolated example, but instead a fairly common mathematical object. It is therefore plausible that some of the polyhedral cages described in this paper could be realised using proteins.

The article is clearly written throughout, and ideally suited to this journal, given the interdisciplinary nature of the topic, and the blend of mathematical sophistication with computational complexity. I fully recommend publication (and do not need to see the article again).

As I was reading the article I spotted some typos and grammatical errors. I list these below for the authors' benefit.

P5 L54 "The edge of the hole-polyhedron specify" -> "The edges of the hole-polyhedron specify"

P5 L56 "When we add a P-gon on the vertex of degree E_h , we must join E_h of its edges, to neighbour faces,..." delete the second comma in this sentence

P6 L31 From a bionanotechnology point of view, the most relevant p-cages, are the ones..." delete the second comma in this sentence (a comma should not be placed between subject and verb)

P6 L38 "Moreover, the connectivity-invariance of the p-cage implies that the hole polyhedra is vertex-transitive or in other words a Cayley graph." Change "polyhedra" to "polyhedron" ("polyhedra" is plural but "is" is 3rd person singular).

P5 L43 "the faces that shares one edge with it": Change "shares" to "share" ("faces" is plural but "shares" is first person singular)

P8 L57 "the rotation symmetry around the axis going through a n-gonal face is a rotation of π/n and not $2\pi/n$ ". Should " π/n " be replaced with " $4\pi/n$ "?

P10 L35 "edges shared with other p-cage face" replace "face" with "faces"

P12 L42 "the vertex of the p-cage are independent parameters." Change "vertex" to "vertices" ("vertex" is singular but "are" is second person plural).

P12 Eq (3.2) notation in the formula for O is slightly imprecise: $n_{\{f,i\}}$ is defined only for i in the range $[0, P-1]$, but the sum is from 0 to N_N-1 . Perhaps some additional notation is needed? Or perhaps the sum should be over both f and i (taking care to count each vertex only once)?

P18 L47 "We have then enumerated all the distribution of hole edges" Change "distribution" to "distributions"

Decision letter (RSPA-2021-0679.R0)

14-Feb-2022

Dear Professor Piette,

On behalf of the Editor, I am pleased to inform you that your Manuscript RSPA-2021-0679 entitled "Characterisation of Near-miss Connectivity-Invariant Homogeneous Convex Polyhedral Cages" has been accepted for publication subject to minor revisions in Proceedings A. Please find the referees' comments below.

The reviewer(s) have recommended publication, but also suggest some minor revisions to your manuscript. Therefore, I invite you to respond to the reviewer(s)' comments and revise your

manuscript. Please note that we have a strict upper limit of 28 pages for each paper. Please endeavour to incorporate any revisions while keeping the paper within journal limits. Please note that page charges are made on all papers longer than 20 pages. If you cannot pay these charges you must reduce your paper to 20 pages before submitting your revision. Your paper has been ESTIMATED to be 20 pages. We cannot proceed with typesetting your paper without your agreement to meet page charges in full should the paper exceed 20 pages when typeset. If you have any questions, please do get in touch.

It is a condition of publication that you submit the revised version of your manuscript within 7 days. If you do not think you will be able to meet this date please let me know in advance of the due date.

To revise your manuscript, log into <https://mc.manuscriptcentral.com/prsa> and enter your Author Centre, where you will find your manuscript title listed under "Manuscripts with Decisions." Under "Actions," click on "Create a Revision." Your manuscript number has been appended to denote a revision.

You will be unable to make your revisions on the originally submitted version of the manuscript. Instead, revise your manuscript and upload a new version through your Author Centre.

When submitting your revised manuscript, you will be able to respond to the comments made by the referee(s) and upload a file "Response to Referees" in Step 1: "View and Respond to Decision Letter". Please provide a point-by-point response to the comments raised by the reviewers and the editor(s). A thorough response to these points will help us to assess your revision quickly. You can also upload a 'tracked changes' version either as part of the 'Response to reviews' or as a 'Main document'.

IMPORTANT: Your original files are available to you when you upload your revised manuscript. Please delete any redundant files before completing the submission process.

When uploading your revised files, please make sure that you include the following as we cannot proceed without these:

- 1) A text file of the manuscript (doc, txt, rtf or tex), including the references, tables (including captions) and figure captions. Please remove any tracked changes from the text before submission. PDF files are not an accepted format for the "Main Document".
- 2) A separate electronic file of each figure (tif, eps or print-quality pdf preferred). The format should be produced directly from original creation package, or original software format.
- 3) Electronic Supplementary Material (ESM): all supplementary materials accompanying an accepted article will be treated as in their final form. Note that the Royal Society will not edit or typeset supplementary material and it will be hosted as provided. Please ensure that the supplementary material includes the paper details where possible (authors, article title, journal name). Supplementary files will be published alongside the paper on the journal website and posted on the online figshare repository (<https://figshare.com>). The heading and legend provided for each supplementary file during the submission process will be used to create the figshare page, so please ensure these are accurate and informative so that your files can be found in searches. Files on figshare will be made available approximately one week before the accompanying article so that the supplementary material can be attributed a unique DOI. Alternatively you may upload a zip folder containing all source files for your manuscript as described above with a PDF as your "Main Document". This should be the full paper as it appears when compiled from the individual files supplied in the zip folder.

Article Funder

Please ensure you fill in the Article Funder question on page 2 to ensure the correct data is collected for FundRef (<http://www.crossref.org/fundref/>).

Media summary

Please ensure you include a short non-technical summary (up to 100 words) of the key findings/importance of your paper. This will be used for to promote your work and marketing purposes (e.g. press releases). The summary should be prepared using the following guidelines:

*Write simple English: this is intended for the general public. Please explain any essential technical terms in a short and simple manner.

*Describe (a) the study (b) its key findings and (c) its implications.

*State why this work is newsworthy, be concise and do not overstate (true 'breakthroughs' are a rarity).

*Ensure that you include valid contact details for the lead author (institutional address, email address, telephone number).

Cover images

We welcome submissions of images for possible use on the cover of Proceedings A. Images should be square in dimension and please ensure that you obtain all relevant copyright permissions before submitting the image to us. If you would like to submit an image for consideration please send your image to proceedingsa@royalsociety.org

Open Access

You are invited to opt for open access, our author pays publishing model. Payment of open access fees will enable your article to be made freely available via the Royal Society website as soon as it is ready for publication. For more information about open access please visit <https://royalsociety.org/journals/authors/open-access/>. The open access fee for this journal is £1700/\$2380/€2040 per article. VAT will be charged where applicable. Please note that if the corresponding author is at an institution that is part of a Read and Publishing deal you are required to select this option. See <https://royalsociety.org/journals/librarians/purchasing/read-and-publish/read-publish-agreements/> for further details.

Once again, thank you for submitting your manuscript to Proceedings A and I look forward to receiving your revision. If you have any questions at all, please do not hesitate to get in touch.

Best wishes

Raminder Shergill

proceedingsa@royalsociety.org

Proceedings A

on behalf of

Dr Andy Sutherland

Board Member

Proceedings A

Reviewer(s)' Comments to Author:

Referee: 1

Comments to the Author(s)

The changes made by the authors before resubmission contributed to improve the manuscript: the numerical efforts and results of identifying near-miss p-cages are now better emphasized. There is a single question about some change in the data: the original manuscript mentions 5160 potential configurations for possible candidates of near-miss p-cages, which is raised to 5743 in the revised ms - where did the increment come from?

The answers given to the referee's questions are mainly acceptable; however, there are still some concerns with Section 1(c) as far as the completeness of theory of regular invariant p-cage graphs are concerned. Two objections which has to be resolved before publication:

P6 Figure 4: even if it is redrawn, 'x=a, y=b,...' - remains in wrong position (should pertain to triangle #2, not #3)

P5 Figure 3: if some hexagons in Archimedean solids can have 'bcbcb', why bases of prisms cannot? Any pattern 'bcb...bc' for 2n-sided prisms (completed by 'a's accordingly) would be a connectivity-invariant configuration, too (even if they are perhaps not metrically realizable). Does it make a change in the catalog of regular connectivity-invariant p-cages?

Typesetting:

P4 l36: p-cage geometries (also P10 l21: p-cage configurations)

References [17]: Möller instead of Møller

The reviewer apologizes for his earlier mistaken comment about P17, Table 11: the question should have been asked as why a P12_2_3_4 arrangement (giving a truncated icosidodecahedron as hole-polyhedron) is not part of the regular p-cages. Explicit exclusion of the above-mentioned Archimedean solid makes the question irrelevant, though.

Referee: 2

Comments to the Author(s)

This article presents the results of a numerical study of polyhedral cages whose faces are almost regular. It is motivated by the recent discovery of a protein cage with these properties. The investigation employs a combination of combinatorial and numerical methods.

To each polyhedral cage the authors associate a dual graph. If the polyhedral cage is connectivity-invariant then the associated graph will be planar and vertex-transitive, and such graphs have been classified by Maschke. The authors proceed to give a combinatorial classification of the possible homogeneous polyhedral cages corresponding to Maschke's graphs. They then use a numerical method to find almost-regular geometric polyhedral cages corresponding to the abstract polyhedral cages in their classification. Their combinatorial classification is clear and well-explained, and their numerical strategy seems well-adapted to the task. In particular, by using a metropolis algorithm they can be reasonably confident of having identified the majority of almost-regular polyhedral cages within their search parameters.

The end result is that a surprisingly large number of almost regular polyhedral cages have been found. So the experimentally-discovered protein cage is not an isolated example, but instead a fairly common mathematical object. It is therefore plausible that some of the polyhedral cages described in this paper could be realised using proteins.

The article is clearly written throughout, and ideally suited to this journal, given the interdisciplinary nature of the topic, and the blend of mathematical sophistication with

computational complexity. I fully recommend publication (and do not need to see the article again).

As I was reading the article I spotted some typos and grammatical errors. I list these below for the authors' benefit.

P5 L54 "The edge of the hole-polyhedron specify" -> "The edges of the hole-polyhedron specify"

P5 L56 "When we add a P-gon on the vertex of degree E_h , we must join E_h of its edges, to neighbour faces,..." delete the second comma in this sentence

P6 L31 From a bionanotechnology point of view, the most relevant p-cages, are the ones..." delete the second comma in this sentence (a comma should not be placed between subject and verb)

P6 L38 "Moreover, the connectivity-invariance of the p-cage implies that the hole polyhedra is vertex-transitive or in other words a Cayley graph." Change "polyhedra" to "polyhedron" ("polyhedra" is plural but "is" is 3rd person singular).

P5 L43 "the faces that shares one edge with it": Change "shares" to "share" ("faces" is plural but "shares" is first person singular)

P8 L57 "the rotation symmetry around the axis going through a n-gonal face is a rotation of π/n and not $2\pi/n$ ". Should " π/n " be replaced with " $4\pi/n$ "?

P10 L35 "edges shared with other p-cage face" replace "face" with "faces"

P12 L42 "the vertex of the p-cage are independent parameters." Change "vertex" to "vertices" ("vertex" is singular but "are" is second person plural).

P12 Eq (3.2) notation in the formula for O is slightly imprecise: $n_{\{f,i\}}$ is defined only for i in the range $[0, P-1]$, but the sum is from 0 to N_N-1 . Perhaps some additional notation is needed? Or perhaps the sum should be over both f and i (taking care to count each vertex only once)?

P18 L47 "We have then enumerated all the distribution of hole edges" Change "distribution" to "distributions"

Decision letter (RSPA-2021-0679.R1)

28-Feb-2022

Dear Professor Piette

I am pleased to inform you that your manuscript entitled "Characterisation of Near-miss Connectivity-Invariant Homogeneous Convex Polyhedral Cages" has been accepted in its final form for publication in Proceedings A.

Our Production Office will be in contact with you in due course. You can expect to receive a proof of your article soon. Please contact the office to let us know if you are likely to be away from e-mail in the near future. If you do not notify us and comments are not received within 5 days of sending the proof, we may publish the paper as it stands.

As a reminder, you have provided the following 'Data accessibility statement' (if applicable). Please remember to make any data sets live prior to publication, and update any links as needed when you receive a proof to check. It is good practice to also add data sets to your reference list. Statement (if applicable): The C++ and Python programs used to generate all the data are available from https://figshare.com/articles/software/Near-miss_Polyhedral_Cages/14061782
doi:10.6084/m9.figshare.14061782

Under the terms of our licence to publish you may post the author generated postprint (ie. your accepted version not the final typeset version) of your manuscript at any time and this can be made freely available. Postprints can be deposited on a personal or institutional website, or a recognised server/repository. Please note however, that the reporting of postprints is subject to a media embargo, and that the status the manuscript should be made clear. Upon publication of the definitive version on the publisher's site, full details and a link should be added.

You can cite the article in advance of publication using its DOI. The DOI will take the form: 10.1098/rspa.XXXX.YYYY, where XXXX and YYYY are the last 8 digits of your manuscript number (eg. if your manuscript number is RSPA-2017-1234 the DOI would be 10.1098/rspa.2017.1234).

For tips on promoting your accepted paper see our blog post:
<https://royalsociety.org/blog/2020/07/promoting-your-latest-paper-and-tracking-your-results/>

On behalf of the Editor of Proceedings A, we look forward to your continued contributions to the Journal.

Sincerely,
Raminder Shergill
proceedingsa@royalsociety.org

Appendix A

Dear Ms Shergill,

As was suggested by yourself and the referee, we have merged our two original papers together, mostly by including the key descriptions of the first paper into what was the second paper (the one we are resubmitting).

To obtain a paper that fits within the journal page limits, we have done the following: for the characterisation of the invariant graph we refer to a paper by H. Maschke. The distribution of the holes edges on the hole-polyhedra graph is derived in the paper except for the graphical proof for the octahedron, dodecahedron and icosahedron which were too long to fit into the paper and which we have included in the supplementary material. The derivation of the geometry of the regular p-cage has also been moved to the supplementary material as they have been obtained by the same method as the near-miss p-cages and this is just an alternative way of obtaining them geometrically.

We believe that we have taken all the comments made by the referee into account and do hope that you will now find the paper suitable for publication in the Royal Society Proceedings. We would also like to thank the referee for their comments and suggestions which helped us improve our manuscript.

Best wishes,

Bernard Piette, Agnieszka Kowalczyk and Jonathan Heddle

We include below our answers to the comments made by the referee for the 2 papers we have originally submitted:

PAPER 1:

#1 General comments

The main goal of the ms is to give a theoretical classification of 'regular invariant p-cages'. Unfortunately, the approach itself is inconsistent. There is a mismatch between assumptions and methodology: invariant p-cages are defined in ms 1 as having indistinguishable faces (i.e., polygons with the same geometry of connections).

We have clarified the definition in the new manuscript. We are also using the term “connectivity-invariant” term instead of “invariant”.

This definition would only imply L_h -regularity of the graph of the hole polyhedra, and would still allow for a p-cage obtained from merging two truncated cubes to be considered invariant. However, the sentence on p5, l37 ('the vertices of the hole-polyhedron must belong to the same number of faces of the same type') and the following formula (3.6) expresses a stronger (symmetry) condition that does not follow from the preceding assumptions (perhaps invariance includes at most L_h geometrically different holes: Figure 3 and so on seems to support this interpretation but it is not declared explicitly).

We have changed notation and use \$E_h\$ instead of \$L_h\$ to avoid ambiguity with \$L_f.E_h\$ regularity is not sufficient as one must also distribute hole-edges. Any isomorphism must map face nodes onto face nodes. We have made the definition more explicit in the new manuscript.

If the number of hole edges of each face around a given hole might be different (as suggested by figure 2d), why should this kind of symmetry constraint be imposed on the face structure of a hole polyhedron?

The connectivity-invariance is imposed to impose some regularity of the p-cage. This is motivated by bionanotechnology where the connected edges are actually protein building blocks connected by specific interactions between amino acids. As protein cages are typically made from multiple identical protein building blocks these interactions cannot be varied between building blocks and are always equivalent where they occur.

Well in the middle of the text, on p11, around line 28, there is an implicit exclusion of mirror symmetry from invariance, another important condition not mentioned among initial assumptions.

We have now introduced and motivated that definition earlier in the manuscript.

These two conditions, however, automatically imply vertex-congruence for the graph of hole polyhedra and enforce the topological symmetry of Archimedean solids (including Platonic solids, prisms and antiprisms) to appear for the same graphs (and hence there is no need either of the numeric search for hole polyhedra or of Appendix A).

We have indeed found that this was proved by H. Maschke in 1896 and we are now referring to that paper.

More about appendices: Appendix B is unnecessarily complicated. (A pair of tangents to one P-gonal face have the same length but it strictly monotonically decreases with the angle they make -> three equal lengths mean equal number of hole edges between each pair of tangents.)

This is indeed much simpler and we would like to thank the referee for the tip.

Figures 3-9 already are all based on the assumption that p-cages have metrically identical holes, so simple symmetry arguments would lead to the same conclusions on the possible repartition scheme.

This is indeed true for the regular cages, but we need this classification for the nearly-regular p-cages as well. Now that we have a single paper, this is clearer.

At this point, the authors present some ideas of generating families of invariant p-cages and exclude some Archimedean polyhedral graphs from the set of possible hole polyhedra but this could have been shown again by much simpler and uniform symmetry arguments. It is strange that 'Platonic Group' appears only among keywords but not in the text of the ms.

It is clearer in the new manuscript that as we are interested in nearly-regular p-cages, that we can't use the symmetry argument to simplify the construction. We are not even requiring that all the faces are identical. With the exception of the regular p-cages, most of the p-cage we have constructed do not have any Platonic sub-group symmetry, only the hole-polyhedra graphs have such symmetries (at graph level, not as isometries).

In summary, the task of searching for 'regular invariant p-cages' seems tacitly been converted into a search of 'regular p-cages with symmetry and connectivity of Archimedean (incl. Platonic, etc.) polyhedra'. This idea does not deserve to be expanded over more than 20 pages but - after a reasonable compactification - can be a section of a single paper dealing with regular and nearly-regular p-cages.

This is indeed what we have now done.

An alternative could have been a systematic but much more general scan (based only on L_h -regularity of graphs of hole polyhedra) for invariant p-cages but it is much more difficult to prove general statements.

We have proved by construction that all the connectivity-invariant p-cages are the one we have studied. There are too many E_h regular graphs to scan all the possible non connectivity-invariant p-cages, even for small graphs and polygons. In any case, from a bio-technology point of view the connectivity-invariant p-cages are the most relevant ones.

#2 Technical remarks

The ms has not been proof-read with care before submission. In addition to inconsistencies in theory that were addressed above, figures and many formulae are typeset without care (examples are listed below among miscellaneous small comments; 'Pn lm' is a reference to 'line m' on 'page n' (in black box)). In addition, the relatively short list of references would have been deserved a proof reading to avoid items looking like "H.S.M. C,..., JCP M"; would it be "H. S. M. Coxeter, M. S. Longuet-Higgins, J. C. P. Miller"?

This has been fixed.

On the other hand, graphics provided by Antiprism are nice and helpful in understanding.

P2 I45: What kind of prisms?

We are only referring to regular prism and we have made this clearer in the manuscript.

P3 I34: What kind of "transformation onto itself"? Indistinguishable = congruent?

We have made this clearer by describing the transformation as an automorphism of the p-cage that maps a face onto another one by mapping hole edge into hole edges (and so linking edges into linking edges).

P3 I37: What if a p-cage is not homogeneous? It is better to state in the same sentence that homogeneous cases are dealt with only.

We have now made it clear we are only interested in homogeneous p-cages.

P3 I43: "a regular p-cages" - saturation no. = no. of adjacent faces if it is saturated? Saturation itself is irrelevant in the ms further on.

We have removed the definition of saturation.

P4 at the bottom: "very substantially"??

We have modified that sentence.

P5 I38: v_i misleading; then (3.8) follows immediately from (3.6) without (3.7)

We have removed that section altogether in the manuscript.

P5 I-1: tetrahedron from 18-gons? Is it excluded somewhere?

This was the Pte_P18_5_5_5 which we don't include in the paper as, for consistency, we now restrict ourselves to polygon with up to 17 edges. This is described in the supplementary material.

Q: is it possible to have different hole polyhedra just by changing the order of successive types of faces? E.g., fig. 2c is 3,4,3,5 but could it be 3,3,4,5?

This is the point of the classification of the distribution of the edge hole. One has to find all the distributions that are compatible with the connectivity-invariance. Most of them leads to p-cages with large deformations.

Invariant p-cage = vertex-congruent p-cage?

The regular p-cages have congruent faces, but the faces of non regular p-cages are not congruent in general. We have explicitly avoided to make that assumption and our results proves us right as most non regular p-cages and faces that are not congruent. We have made this points more explicitly in the manuscript.

P7 (3.9): isn't $q = U$?

Indeed! We would like to thank the referee for spotting this typo.

P9 Figure 4: 'x=a, y=b,...' - typeset in wrong place (should pertain to panel 2)

We have redrawn that figure.

In general, it is not clear what the condition of invariance is in Figs. 3-9: the term 'nontrivial invariant configuration' has rather to do something with lower order of symmetry but anything is invariant where the vertices have the same order of labels (e.g., 'abc'). If some hexagons in Archimedean

solids can have 'bcbcbc', why bases of prisms cannot? Statements in this section are not clear and proofs are even less.

We have improved the description of the connectivity-invariance that we have to impose.

P11 I38: It is true only if invariance excludes reflection AND the hole polyhedron is forced to be Archimedean.

This is indeed what H. Maschke has proved and we explicitly exclude invariance that requires a reflection.

P12 I38: what is the definition for 'dressed'?

We have now refrained from using that expression.

P13 I49: '3elQ' ?

We would like to thank the referee for spotting this typo.

P14 (4.6) uses χ with reference to Fig. 10 but there is no χ .

The problem was that our drawing package, inkscape, uses a 'chi' which look very much like an 'X'. We have redrawn it to make the difference more obvious.

P15 Table 7: the last column lists antiprisms, not equivalent prisms. There is also a mismatch of P in the last row.

As was explained in the text, all the p-cages constructed from antiprism are degenerate: some of the hole-edges are merged with hole-edges of other faces and as such form p-cages which are identical to p-cage obtained from a different solid. These solids were listed in the last column. As these p-cages are not really new, we have removed them from the manuscript and only list them in the supplementary material.

P16 Table 9: 'sub-platonic' is not defined; terms from 'Other name' are partially missing but mostly irrelevant;

We have rephrased that table title.

I43: 'vertices' instead of 'edges'

We would like to thank the referee for spotting this mistake.

P17 Table 11: Why 20 6-tuples of coplanar 6k-gons are disregarded as a truncated icosahedral p-cage if other non-strictly convex cages are allowed (Ati_P12_1_1_1_1_3, Ati_P18_2_2_2_2_5)? Similarly for truncated tetrahedra and octahedra...

The vertices of the truncated icosahedron have only 3 edges attached to them and so one can only have 3 holes around them, not 6.

P15 l47-53: The proof is extremely sloppy and full of typos. What is v_1 exactly? '...and so ... $\gamma = \pi/6$ ' - why? Three rows below: the same angle between v_1 and l_3 is $\zeta(\pi - \eta)/2$ but neither η nor ζ is defined earlier. In the following formula γ is rather $\cos \gamma$, finally, $\gamma = Q\pi$ is surely incorrect. In general, Fig. 12 gives not many help in understanding - what are the dotted green lines outside tents?

We have rewritten the proof differently. This section is now in the supplementary material.

ref. 1,2: "XXXea = XXX et al."? (2x)

We would like to thank the referee for spotting this typo.

PAPER 2:

The main idea is the sufficient distortion of hole edges to let shared edges of p-cages be tuned according to the conditions of continuity, planarity and convexity is elegant.

However, there are still major concerns about the applied optimization method.

It is not clear what kind of connectivity is assumed for the initial configurations. Final result must sensibly depend on topological conditions.

We have made this point clearer in the manuscript . First of all, we are using the word “connectivity invariant” instead if “invariant” to emphasise the fact that the invariance is at graph level and not metric. We have used the following description in the text:

A homogeneous p-cage is said to be connectivity-invariant if all the faces are indistinguishable from their connectivity, or in other words if for any pair of faces A and B there is an automorphism of the p-cage graph onto itself that maps the vertices of face A onto the vertices of face B, such that the connecting edges are mapped to connecting edges and so that hole edges are also mapped to hole edges. The faces are assumed to be isomorphic but not isometric. For nano-bio-technological motivations, in this paper we do not consider the p-cages for which one must involve a reflection to achieve the connectivity-invariance.

The optimization procedure looks rather heuristic: why $c_l + c_a$ is adopted specifically? Relative equivalence of perturbations of leg lengths and right angles in an isosceles right triangle is credibly shown in the paper but angular deviation of adjacent edges of, e.g., a 17-gon is $2\pi/17 <$

This is indeed heuristic. Ultimately we obtain a full parameter family of near-miss p-cage so the actual choice does not matter much. The angle we use is α which is $2\pi(1-1/P)$, not $\beta=2\pi/P$.

As a summary of all above, a 'nearly fatal' misconception is explicitly declared at the end of the paper (page 15, lines 35-37). Some of the near-miss cages are PROVEN to have symmetry (see [1]) but the iterative method leaves this property unrevealed. Although there is no evidence to happen so always but it seems to be a strong conjecture that MOST p-cages shown here has the same level of metric symmetry as the symmetry of its connectivity graph is. It is even strengthened by the aim of dealing with 'invariant' cages only (it is true, however, that invariance is not a direct proof of a metric vertex-transitivity of near-miss cages). It can also be concluded that a much more efficient iteration could have been programmed based on a few geometric parameters using symmetry arguments.

The symmetry of the near-miss p-cage is only at graph level. We explicitly decided not to assume that the faces are all identical as there is no reason to assume that they should be. This was eventually confirmed by the results we obtained (as described in detail in the description of all the p-cages in the supplementary material). This is not due to an inaccuracy of the minimisation method. We have repeated the optimisation a number of times for a range of p-cages and found the “optimal” cages to always be the same. Moreover, to evaluate the accuracy of our minimisation procedure, we have applied it to the regular cage and found the error to be $1e-6$. This is much smaller than the difference in size that we observe between the faces of a given p-cage.

One could indeed assume that all the faces are identical and then assume some p -cage symmetry. This would allow one to reduce the number of degrees of freedom considerably, but the resulting p -cages would have larger deformations.

A possible idea is as follows: look at the name code of each near-miss p -cage and depart from the symmetry group of order n it belongs to. Any plane H can be given in 3D by 3 scalars (this number can perhaps be reduced by considerations in elementary geometry, especially for regular p -cages). In the following step, based on all n images of H under symmetry operations, lines of intersection and even the midpoint of shared edge can be found (analytically?) between connected faces (if no shared edge is assumed, no intersection is considered). If the hole polyhedron is L_h -regular, intersections define a set of n copies of L_h -gons. In the last step, 'inscribed' irregular P -gons are fitted to L_h -gons (respecting L_h edge midpoints as well) for different values of P . In this way, symmetry and planarity will never be violated, and the method possibly offers an iteration in much less variables.

As explained above, we did not want to assume such symmetry of the faces.

#2 Technical remarks

The notation of the m s is positively sloppy: it starts, e.g., by denoting the vertex number of polygons by P then converts to N gradually in Section 3 without giving any notice. Symbols for cyclic subscripts are inconsistent, formulae frequently miss some letters or use some of them incorrectly. Some details and suggestions are given in the following list.

N refers to the number of p -cage faces while P is the number of edges of the polygonal faces.

P4 Fig. 1b It is better to have the same number of edges as in 1a, i.e., 5. The exact meaning of ϵ_i cannot be seen in part b.

We have redrawn that figure to make it clearer.

P4 I47: LHS of (2.1) should be duplicated

We have made the sum over the index i explicit.

P4 I51: It would be better to introduce q before (2.1)

We don't really see why as q is not used in eq (2.1)

P5 l11: The equation is OK only if β_{i0} (and not β_i) is $2\pi/P$. It should be stated explicitly what do α_{i0} , β_{i0} , γ_{i0} stand for. Note that if ' $i0$ ' refers to the original (regular) configuration, then κ_i and $\kappa_{\alpha,i}$ must have opposite sign, since α and γ cannot increase at the same time.

We have rewritten this section to make it clearer. κ_{α} and κ have indeed opposite sign and we have corrected it.

P5 (2.6) π instead of β_i (or 2 instead of P , cf. (2.3))

We have corrected eq 2.6

P5 (2.7) Missing factor 2 on the RHS

We have corrected eq 2.7

P5 l31: Isn't it 'sum of γ other edge lengths' of the same sub-faces hole'?

The condition is correct as it is .

P5 l35: Equality can only hold if all γ s are 0 (provided A is understood as shown in Fig. 1c)

Indeed, this is what was stated on line 35.

P5 l40-41: why not ' A_j ' on the left hand side (A_i is on the RHS)?

We would like to thank the referee for spotting this typo.

P5 l49: 'majorate or minorate': isn't it governed by the sign of κ_i ??

This is equivalent as $1/(1+k)$ is approximately $1-k$ for small k . With our choice of parametrisation k is the deformation as a fraction of the reference length.

P5 (2.10) is true only if shared edges of P -gons are of unit length (but L_f is only defined later on P7 l31). Moreover, (2.10) and (2.11) should be merged by summing terms from $i=0$ to q_k-1 .

We have introduced a length scale in these 2 expression. It cancels out in what was (2.12).

Equation (2.10) and (2.11), they describe 2 different types of edge hole and their sum is performed in what was equation (2.12)

Generally speaking, 'deformations' are first mentioned in this section on P5, l10. It would greatly help understanding if a clear statement was made here what kind of deformations are considered (it seems that unit edge length of P -gons is kept and A_i is changed but it is not written explicitly). A figure should be attached as well to show the meaning of A_i , angles with subscript ' i ' and ' $i0$ ', all magnification factors κ , etc.

We have modified this section to make clearer, but we are too short of space to add an extra figure.

P6 I35: Isn't strict planarity required? It is not subject then to minimization...

We need to impose planarity as a constraint because the vertices of the p-cage faces are the degree of freedom. This makes the algorithm computationally tractable. One could use the faces as degree of freedom, but this would make the algorithm much slower because of all the algebraic constraints one would have to solve. We have added a comment in the text to motivate our approach.

P6 I38: The definition of N_N is a bit unclear: do nodes count twice if they are on a shared edge? (3.2) could make it clear if the summation was expanded (however, this is not a relevant question as far as the cage is 'sufficiently' symmetric). Similarly, in (3.3) and (3.7), $\sum_{i=1}^P$ (or $\sum_{i=0}^{P-1}$) is better instead of 'face nodes'.

We have made it explicit that the p-cage nodes are counted only once.

We have used explicit indices for the 2 sums (3.3) and (3.7).

P6-7: There is a mismatch between (3.1) and (3.6): $(i+1)/N_f \ll i/N_{f+1}$ (and their ranges $0..N_f-1$ and $1..N_f$ are also different). N_f could be P_f according to the initial notation.

We have corrected the notation using indices ranging from 0 to $N-1$ to make the modulo operator work correctly.

P7 I26: 'optimize the symmetry of the cage'??

We have improved that description.

P7 I43: 'isosceles rectangular triangle'  '...right triangle'?

This should have been "isosceles right triangle". Now corrected.

P7 I56 'in radians'

We would like to thank the referee for spotting this typo.

P8 (3.14) seems to penalize convexity, not concavity

Indeed, we have corrected it by changing the sign of the argument in the Heaviside function.

P8 I31: Is $d = M-F$? Please give a definition. If the present condition does not strictly impose convexity, why not to consider $H((W_i \times W_i) \cdot s)$?

This was defined page 6 but we are now using an equation number to make it clearer.

P9 I7: (d)  (3.18); 'where'  'were'

We would like to thank the referee for spotting this typo.

P9, l14 as well as l41: 'edge lengths'

We would like to thank the referee for spotting this spelling mistake.

P9 (3.19) is only necessary to get separate limits (both set later on to 0.1) for Δ_l and Δ_a . It would have been simpler and more straightforward to constrain $(\Delta_l^2 + \Delta_a^2)$, directly following from the functional instead of $|\Delta_l| + |\Delta_a|$.

We could have done what the referee suggest, but not knowing what the results would be we wanted to find p-cages with both small $|\Delta_l|$ and small $|\Delta_a|$ independently. For some cages it turns out that the $|\Delta_a|$ is effectively 0 while $|\Delta_l|$ can vary. For others both the edge length and angles must be deformed.

P9 (3.20) uses a misleading (although not incorrect) notation: i stood earlier on for individual edges of a face f , A_i denoted a diagonal ('edge of sub-faces hole'), not an angle: this later was $\alpha_{f,i}$ as defined in (3.1). $i,j \rightarrow f,g$, as well as $k \rightarrow i$ would be more consequent. P and N within the same formula??!

We have modified this equation to make it more clearer.

The text preceding (3.20) would be easier to follow if its motivation was given: since those p-cages may exhibit some topological symmetry, they are expected to retain some metric symmetry as well. In order to quantify the differences properly, a preceding identification of 'nearly similar' faces should be performed: one face is unchanged, another is successively rotated until a 'quasi-overlapping' is detected. It should explicitly be written in the text.

We have rephrased this section to motivate it and make it clearer.

P9 l57: 'tiling'

We would like to thank the referee for spotting this spelling mistake.

P10, Figure 2: all p-cages listed here exhibit symmetries of some Archimedean solids (in a general sense, including those of prisms, antiprisms and even of Platonic solids as well): how can some of them have notable differences between faces (Ω_a and Ω_l both exceed 1% in $A_{P7_1_1_2}$)? What kind of terminal condition for the iteration was used?

The deformations are very small and so the p-cages looks nearly symmetric, but in fact they aren't. This is somehow one of their properties that makes them so interesting.

P12 l50: $A_{2_4_4}$ must be $A_{P13_2_4_4}$

We would like to thank the referee for spotting that typo.

P13 l58: 'all the possible cage connectivities': it is not proved anywhere.

This is by construction. We have rephrased it to make it clearer.